**Effect of land use legacy on the future carbon sink for the conterminous U.S.**
Benjamin S.Felzer
Earth and Environmental Sciences
Lehigh University
1 W. Packer Ave.
Bethlehem, PA 1807
bsf208@lehigh.edu

**Abstract**

Modeling the effects of the terrestrial carbon sink in the future depends upon not just current-day land use and land cover (LULC), but also the legacy of past LULC change (LULCC), which is often not considered. The age distribution of trees in the forest depends upon the history of past disturbances, while the nutrients in the soil depend upon past LULC. Thus establishing the correct initial state of the vegetation and soil is crucial to model accurately the effect of biogeochemical cycling with environmental change in the future. This study models the effects of LULCC from 1750 to 2014 using the Land Use Harmonization dataset (LUH2) of land use transitions with the Terrestrial Ecosystems Model (TEM) for the conterminous U.S. Modeled LULC include plant functional types (PFTs) of potential vegetation, as well as managed cropland, pastureland, and urban areas. LULCC is treated using a cohort approach, in which a separate cohort occurs every year there is a land use transition, thereby ensuring proper age structure of forests and regrowth with the correct soil nutrients. From 2000-2014 the modeled Net Ecosystem Productivity (NEP) is 989 TgCyr$^{-1}$ for the conterminous U.S., but only -15 TgCyr$^{-1}$ if accounting for carbon lost from land use transitions and management.

The hypothesis is that the initial state of the vegetation and soils significantly affects the future state of the terrestrial carbon sink. In this study, LULC remains constant in the future, with the NCAR CCSM4 RCP8.5 climate used to force the TEM-Hydro model. The following experiments are run from 2015 to 2100, including a) restarting from existing cohorts in 2014 (RESTART), b) reinitializing in 2015 based on condensing the cohorts for each PFT into a single cohort (CONDENSED), and c) restarting from average cohort conditions for each PFT (AVERAGE). The NEP is too low when using condensed cohorts without reinitializing due to a larger increase in heterotrophic respiration ($R_h$) resulting from the assumption of mature forests. The carbon stocks are larger than using all the cohorts if condensed cohorts are reinitialized due to the assumption of mature, equilibrated forests. Where nitrogen-limited, forest regrowth is enhanced if regrowth starts from more nutrient-rich conditions. Water fluxes are dominated by environmental factors, but can be slightly dependent upon the underlying carbon dynamics. It is therefore necessary to account for past disturbances when modeling future changes in carbon dynamics.

## 1 Introduction

Globally, during the 21$^{st}$ century, land use and land cover change (LULCC) has accounted for 14% of the total anthropogenic carbon emissions (Friedlingstein et al., 2020). LULCC have been responsible for the largest losses of carbon from the land in the conterminous U.S. since the 1700s, with growth enhancements from $CO_2$ fertilization and nitrogen deposition only partially countering this loss since the 1950s (Felzer and Jiang, 2018). Reforestation and afforestation have been the primary drivers for this enhanced sink (Kondo et al., 2018), especially growing back with rising $CO_2$ levels (Strassmann et al., 2008). This paper addresses the question of the role of land legacy in the future carbon sink in conterminous U.S. How inappropriate is it to initialize a model with current-day land use and land cover (LULC) for a 21$^{st}$ century simulation, which avoids the disturbance history and forest recovery from the 20$^{th}$ century and earlier?

Many modeling studies have been conducted to explore the role of LULCC relative to other
environmental factors like $CO_2$ fertilization, N deposition, and ozone both historically and into
the future.  For example, studies have shown LULCC to be the most important cause of reduced
carbon inventory in the future due to loss of forest (Mahowald et al., 2017), while $CO_2$
fertilization increases the sink (Tharammal et al., 2019).  Reforestation, including regrowth from
timber harvest, and avoided deforestation, can increase the carbon sink in the future (Arneth et
al., 2017; Zhao et al., 2013).  Remotely sensed data from 1973-2010 have shown that both
reduced forest area and older forest age have contributed to a reduced C sink in the conterminous
US (Sleeter et al., 2018).  Legacy carbon fluxes from deforestation can be in the form of
emissions from dead biomass, soils, and forest products, or uptake in regrowing secondary
forests (Houghton et al., 2012).
Only a few models (e.g. (Felzer and Jiang, 2018; Shevliakova et al., 2009)) have included forest
demography, to accurately track the effects of disturbance in regrowing forests.  Krause et al.
(2020) showed that including land legacy effects increases future carbon storage as ecosystems
regrow and adapt to higher levels of $CO_2$ and N deposition.  Since ecosystems are not in
equilibration with current-day land use, there will be continued carbon uptake even if climate
change and land use are held constant, due to regrowth from abandoned agriculture and $CO_2$
fertilization (Krause et al., 2019).  Pugh et al. (2019) surmised that there will be a large carbon
sink from regrowth in the future regardless of environmental change as long as current
disturbance rates continue at historical levels.  Lu et al. (2015) found that using corrected Forest
Inventory Analysis (FIA) data (Pan et al., 2011) applied to a dataset of annual land use
transitions (Hurtt et al., 2011) nearly doubled the carbon sink due to younger forests in the
corrected data.  Thom et al. (2018) points out that it is important to develop initial conditions to
account for past disturbance in order to capture the observed state.  This idea is tested in the
current study by determining the difference in future carbon sink between initial conditions that
do capture disturbance since 1750 and reinitialized initial conditions.
Two factors that determine the carbon sink strength of regrowing forests are the stand age
distribution of the trees in the forest and the nutrient levels of the soil.  The age distribution
depends upon the timing and magnitude of past disturbances.  Soil nutrient conditions depend
upon the prior history of land use and management.  Several studies show that forest regrowing
from nutrient-rich fertilized agricultural land exhibit less resilience for climate change but higher
growth rates.  European beech trees on former agricultural land had lower C:N and higher P,
which resulted in less carbon allocation to roots, reducing resilience to drought (Mausolf et al.,
2018).  Similarly, Von Ohemib (2014) found these same changes led to higher tree ring width
due to more litter decomposition and higher N mineralization rates, as well as reduced resiliency.
In terms of Net Ecosystem Productivity (NEP), reforestation sites exhibited reduced NEP due to
loss of carbon from the forest floor or soils during early recovery (Pan et al., 2011) but enhanced
NEP in afforestation sites due to replacement of depleted pools (Post and Kwon, 2000).
This study explores the question of land legacy on the future carbon sink by comparing model
simulations with full forest demography with those based on reinitializing initial conditions to
the present.  The analysis looks at both carbon fluxes and stocks to determine how these vary
regionally and integrated over the entire conterminous U.S. It explores the role of forest stand

age and soil nutrients in determining forest regrowth and tests the hypothesis that it is crucial to capture the effects of historical land legacy in order to accurately model the future carbon sink.

## 2 Methods

This study uses the Terrestrial Ecosystems Model- Hydro Version 2 to explore the role of historical land use legacy (from 1750 to 2014) on future (2014-2099) carbon storage.  The recent LUH2 version of land use transitions (Hurtt et al., 2020) is used to reconstruct the full cohort of LULCC since 1750, while LULC is kept constant for the 21$^{st}$ century.  Three sets of experiments explore the role of fully accounting for past land legacy, reinitializing initial conditions and not accounting for land legacy at all, and initial conditions based on averaging the final state of the full cohorts in 2014 to determine if corrected initial conditions are sufficient.

## 2.1 Model Description

The Terrestrial Ecosystems Model version Hydro (TEM-Hydro – (Felzer, 2012; Felzer et al., 2009; Felzer et al., 2011) is a fully prognostic biogeochemical model of carbon, nitrogen and water dynamics between vegetation and soils.  A complete description of the model can be found in Felzer et al. (2009) (2011) and Felzer (2012).  The model structure is illustrated in summary figures (Fig. S1a) along with how human disturbance is treated, which is relevant to this paper (Fig. S1b).  A cohort approach is developed to convert a dataset of land use transitions (Hurtt et al. (2011; 2020) to annual cohorts of land use and land cover change (Hayes et al., 2011; Lu et al., 2015), whose purpose is to retain the soil characteristics of the cohort from which disturbance occurred and maintain appropriate growth and stand age of newly developed cohorts (Fig. S2a).  This approach involves first using the LUH2 dataset to establish the fractional land cover type at the starting year of 1750.  The primary and secondary vegetation are replaced with their potential vegetation values (as described in Raich et al. (1991)), while other managed lands include croplands, pasturelands, and urban, with the multiple types of crops and pastures combined into single values for each, respectively. Disturbances (including timber harvest) involve the creation of new cohorts, with the corresponding area adjusted from the original cohort.  Therefore, soil nutrients and forest stand age are tracked separately for each disturbance.  The output are then area-weighted for each of the cohorts.  Since this approach tracks each cohort separately, it is possible to end up with thousands of cohorts for a single grid cell by 2014.  A complete description of this approach can be found in Felzer and Jiang (2018).  New to this study is that the initial vegetation is started in 1750 (consistent with Allan et al. (2021) baseline period) and subsequent transitions were determined until 2014 (Fig. S2b, c, d, e) to align with the temporal range of climate datasets.  The result for a single grid cell is usually hundreds of cohorts by the year 2014, accounting for all transitions between primary and secondary vegetation, cropland, pastureland, and urban areas, as well as timber harvest.

The partitioning of disturbance products and fluxes for agriculture and timber harvest and management practices and calibration are described in Felzer and Jiang  (2018).  In this study both croplands and turflawn (urban) are fertilized (using the approach taken in Felzer et al. (2018), while no additional fertilization (beyond that provided by livestock) is applied to pasture.  A few additional modifications were made for this study.  Irrigation was added to arid croplands,

because inorganic nitrogen was accumulating due to lack of leaching.  The same scheme as used in Felzer (2012) for turflawn was applied to croplands receiving less than 200 mm of water per month during the growing season.  The other change applies to abandoned cropland.  Cropland abandoned before there was major chemical fertilization in the 1960s were too nutrient depleted in the model, and the forest regrowth occurred with reduced biomass, so 15 gN/m2/month during the year of disturbance was added following crop abandonment to ensure at least limited forest regrowth.

**2.2 Experimental Design**

Six simulations (Table 1) were designed to determine the effect of land legacy.  The HISTORICAL run applies the full cohorts from 1750 to 2014, allowing for the Hurtt et al. (2020) record of LULCC as described in the Methods.  The HISTCONST run is the HISTORICAL run but with LULC held constant at 2014 value, so includes other effects related to climate, $CO_2$, N deposition and ozone.  Therefore the difference between HISTORICAL ad HISTCONST is the effect of LULCC.  The HISTCOND run is a transient run like HISTORICAL except that the Plant Functional Types (PFTs) are condensed to a single cohort for each PFT in a give year.  It is essentially the fractional land cover per year, so the difference between HISTORICAL and HISTCONST illustrates the effect of forest demography.  Because multiple land-use transitions are incorporated into single cohorts, it is not possible to accurately incorporate the true disturbance, so conversion and product fluxes results from land-use change are not included in this run.  The RESTART run uses restart files from the full suite of cohorts in 2014 to run from 2015 to 2099, keeping LULC constant with the 2014 cohorts.  This run is essentially just a continuation of the HISTORICAL run.  The CONDENSED run reinitializes (i.e. reequilibrates) a condensed version of the cohorts in 2014 to provide initial conditions for the 2015 to 2099 period.  In this run, the 2014 cohorts are condensed to a single cohort for each pft (with primary and secondary of the original PFT tracked separately), with the fractional areas determined based on the 2014 cohorts.  These condensed cohorts are then each reequilibrated at the start.  The TEMRESTART run uses a restart file for 2014 that is based on the average of the restart conditions for each of the cohorts, and then uses the condensed cohorts for the 2015 to 2099 period. Thus the TEMRESTART run uses the same number of cohorts as the CONDENSED run, but does not reequilibrate at the start.  So both the CONDENSED and TEMRESTART runs used the simplified, condensed cohorts, but start with different initial conditions.  The motivation for these two condensed-PFT runs is to reduce computational time by eliminating the need to run potentially thousands of land-use legacy cohorts for each grid when starting from present-day conditions.  The difference between the RESTART and CONDENSED runs shows the effect of including land legacy on future carbon dynamics.  Note that the RESTART run will also incorporate effects of changing climate, $CO_2$, ozone, N deposition and fertilization, which cannot be captured in the CONDENSED run.  The TEMRESTART run shows if it is possible to condense the initial conditions from a full suite of cohorts to produce the same results as the RESTART run.

**Table 1:** Model Experiments

| Experiment | HISTORICAL | HISTCONST | HISTCOND | RESTART | CONDENSED | TEMRESTART |
|---|---|---|---|---|---|---|
| LULCC | Transient | Constant 2014 | Transient | Constant 2014 | Constant 2014 | Constant 2014 |
| Number Cohorts | Transient | Condensed ** | Condensed *** | Full at 2014* | Condensed ** | Condensed* ** |
| Initialization | Equilibrate 1750 | Equilibrate 1750 | Equilibrate 1750 | Continuation of Historical | Equilibrate 2015 | Average from HISTORICAL |
| Time Period | 1750-2014 | 1750-2014 | 1750-2014 | 2015-2099 | 2015-2099 | 2015-2099 |
| $CO_2$ | CRU4.04 | CRU4.04 | CRU4.04 | RCP8.5 | RCP8.5 | RCP8.5 |
| N Deposition | Tian et al. 2010 | Tian et al. 2010 | Tian et al. 2010 | Constant 2014 | Constant 2014 | Constant 2014 |
| N Fertilization (crops/turf) | Felzer et al. 2018 | Felzer et al. 2018 | Felzer et al. 2018 | Constant 2014 | Constant 2014 | Constant 2014 |
| Ozone (AOT40) | Felzer et al. 2004 | Felzer et al. 2004 | Felzer et al. 2004 | Constant 2014 | Constant 2014 | Constant 2014 |

* Maximum cohorts for a grid in 2014 is 1020
** Maximum cohorts for a grid is 7, because primary vegetation is treated separately from
secondary
*** Maximum cohorts for a grid is 5 (e.g. mixed potential vegetation with two cohorts, cropland,
pasture, urban)
The model is run monthly at a spatial resolution of 0.5$^o$ x 0.5$^o$. Input datasets include transient
climate (surface air temperature, diurnal temperature range, precipitation, fractional cloud cover
to derive net irradiance at the surface and photosynthetically active radiation (PAR), vapor
pressure), climatological wind speed (as in (Felzer et al., 2011)), and annual atmospheric $CO_2$
from 1901-2014 based on CRU4.04 (Harris et al., 2014). Gridded transient climate data are not
available prior to 1901, so climate variables from 1750 – 1849 are taken from the MPI-ESM-P
past 1000-year simulation and 1850-1900 from the MPI-ESM-P historical simulation (Schmidt et
al., 2014). The downscaling and bias correction is similar as to what was done in Felzer and
Jiang (Felzer and Jiang, 2018), but starting in 1750 instead of 1700, using the CRU4.04 data
from 1901-1930. The resultant U.S. mean climate from 1750 is shown in Fig. S3. Surface ozone
((Felzer et al., 2004), nitrogen deposition (Tian et al., 2010), and soil texture and elevation
datasets are similar to those used in Felzer et al. (2011).
The future climate data (2015-2099) are taken from the Multivariate Adaptive Constructed
Analogs (MACA) statistically downscaled Coupled Model Intercomparison Project 5 (CMIP5)
data (Abatzoglou and Brown, 2012), using the National Center for Atmospheric Research
(NCAR) Community Climate System Model version 4 (CCSM4) RCP8.5 emissions scenario
(r6i1p1 ensemble). The downscaled resolution is at 4 km but has been extrapolated to the half-
degree TEM grid for this study by averaging over all the 4 km values within the larger half-
degree grid cell. Net irradiance is used instead of clouds for the future data. The TEM cloud
scheme was adjusted for the historical cloud data to bias-correct to ensure continuity of net
irradiance between the historical and future data. The CRU4.04 data does not include irradiance,
which is why it was necessary to use clouds for the historical period, but since net irradiance is
more directly used by the model, that was chosen for the future period. The results (Fig. S3)
show a continuity for climate during the transition between the historical CRU4.04 and future
RCP8.5 in 2014 for all the variables. Future RCP8.5 $CO_2$ data are taken from CMIP5
recommendations (Meinshausen et al., 2011) . The ozone and N deposition values are kept at
their 2014 levels (which are held constant after 2000 for ozone).
The decision to base climate prior to 1900, prior to the gridded historical data, was made to
capture more realistic climate variations during the period from 1750 to 1900, such as the Little
Ice Age (LIA), which lasted through the 19[th] century (Bradley and Jonest, 1993; Mann, 2002).
The temperature record from the MPI-ESM-P model does show signs of temperature climbing
out of a cold peak after 1818 but remaining cool throughout the rest of the century (Figure S3),
which is consistent with Northern Hemisphere proxy records (Mann et al., 2008). Since this
study is for the conterminous U.S., it does not show as strong an LIA signal as would be
expected from records in the North Atlantic. The decision to then use historical CRU4.04
climate rather than modeled climate from 1901-2014 is to more accurately capture the true
interannual variability, which would be entirely lost by using output from a climate model. All
three datasets have been downscaled and bias corrected to produce a seamless record of climate
from 1750-2099.
The model is initially calibrated for specific PFTs without disturbance, though with agricultural
and urban management where necessary, to determine coefficients for the flux equations before
extrapolation to the entire U.S. Note that each experiment is not calibrated individually. The
HISTORICAL run is first equilibrated based on repeated use of the 1750-1779 climate in order
to establish initial conditions of carbon and nitrogen stocks (which are required to numerically
solve the fundamental model equations), and then the transient runs are started from 1750 to
2014. The CONDENSED run is first equilibrated based on repeated use of the 1986-2015
climate from the HISTORICAL run, and the transient runs are from 2015 to 2099. Results of
NEP or Net Carbon Exchange (NCE) fluxes are reported as $TgCyr^{-1}$, while cumulative NCE, a
measure of net carbon accumulation over some time periods, is reported as PgC. NCE is the NEP
plus carbon lost through land-use conversion or by decomposition of agricultural or timber
harvest products. Model input, forcing data and output results are publicly available at
http://go.lehigh.edu/landlegacy.
**3. Results**
The historical (1750-2014) NEP (from HISTORICAL) starts to increase in the 1870s (Fig. 1),
consistent with the time period when $CO_2$ levels start to increase and there is a slight warming,
though there is also a decrease in precipitation during this period (Fig. S3). The separation of
NCE from NEP signifies the results of LULCC, which become more pronounced after the 1850s
when timber harvest begins and pasture and cropland increase at the expense of forest, pasture,
and grassland (Fig. S2a).  The cumulative NEP is 87 PgC, while the cumulative NCE is -42 PgC.
So climate and $CO_2$ conditions cause the land to be a net carbon sink, but LULCC makes the
land a net carbon source.

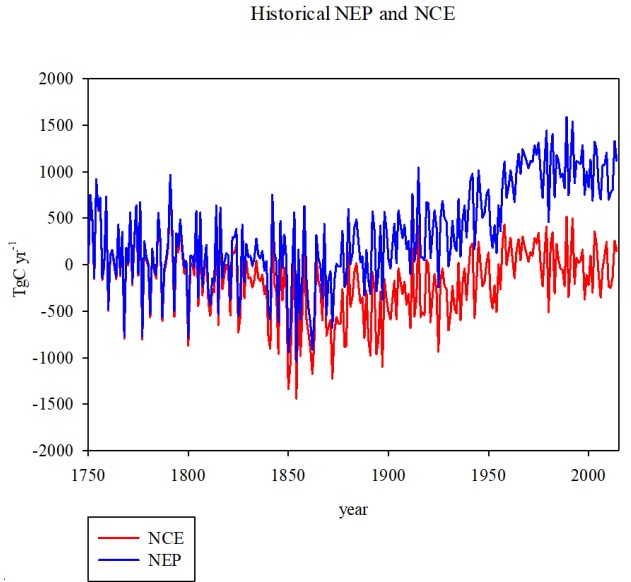


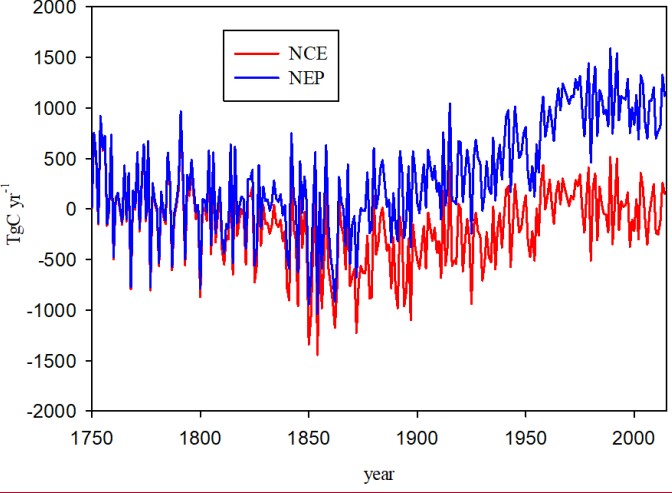


**Figure 1:** Net Carbon Exchange (NCE) and Net Ecosystem Productivity (NEP) for the
HISTORICAL run.  NCE includes fluxes from agricultural conversion and abandonment and
decomposition of agricultural products.
The effect of including LULCC is evident in the difference between HISTORICAL and
HISTCONST (Fig. 2).  While the final cumulative NEP is close by the year 2014, the use of
actual land-use transitions lowers the NEP, especially during the early years, consistent with the
results of Felzer and Jiang (2018) that the effect of deforestation reduces the NEP, while the
larger area of mature forest do not contribute much to positive NEP. The vegetation and soil
carbon start out substantially higher in HISTORICAL, while without LULCC they remain
relatively constant in HISTCONST, which shows the effects of the other environmental changes
like climate, $CO_2$, N deposition, and ozone. The HISTCOND compared to HISTORICAL shows
that the inclusion of forest demography does increase the cumulative NEP by 2014 and lowers
the vegetation and soil carbon estimates, as changing forest area is incorporated into existing
forests rather than separate cohorts to allow for forest regrowth.

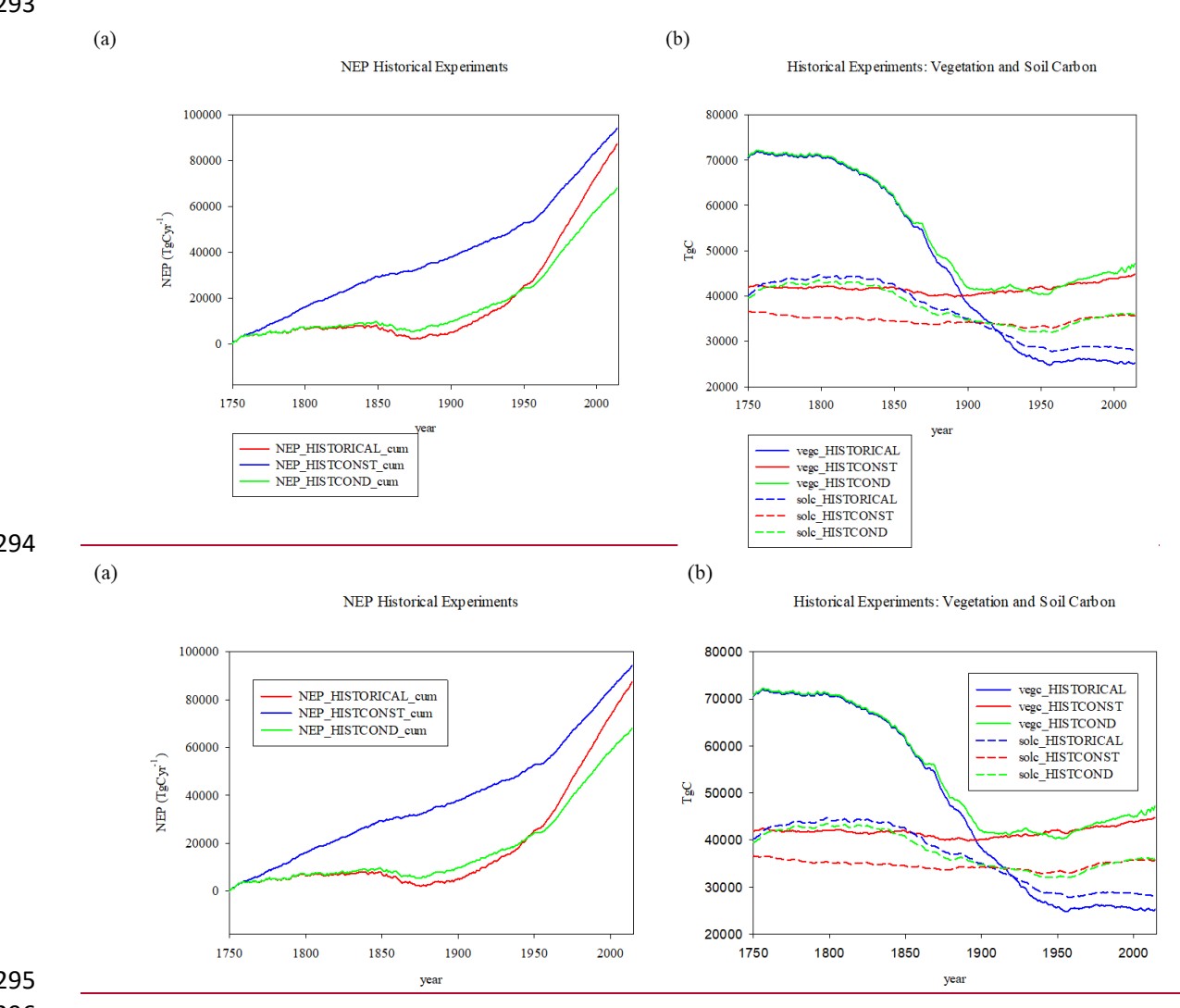

**Figure 2:** Historical experiments a) Cumulative NEP (TgCyr-1), b) Vegetation carbon (vegc)
and soil carbon (solc) in TgC. Experiments are HISTORICAL, HISTCONST, and HISTCOND
(see text).

In the future runs, the RESTART run is considered the "actual" to validate the others against, as
it is the run that includes effects of all the individual cohorts. The CONDENSED run is the
effect of condensing all the cohorts to single PFTs and the TEMRESTART is the result of
averaging the initial conditions for each of the cohorts in 2014. The NEP and NCE of the
CONDENSED is lower than the RESTART and TEMRESTART, especially at the start of the
runs (Fig. 3), because reinitializing each grid is based on the assumption of NEP as close to zero
as possible.  The cumulative result in 2099 is NEP of 76 PgC in the RESTART run, 80 PgC in
the TEMRESTART, and 63 PgC in the CONDENSED.  The cumulative NCE of the RESTART
and TEMRESTART is close beyond the starting years, resulting in 20 and 18 PgC respectively,
while it is lower (9.6 PgC) for the CONDENSED run.  NCE still differs from NEP without
LULCC because of crop decomposition, animal respiration, and crop residue fluxes.  NCE  of
the RESTART and TEMRESTART runs are much lower than NEP of those runs because of
product decomposition left over from the HISTORICAL run.  By the end of the century there is
no significant differences in the annual carbon fluxes, but the condensed run has significantly
lower cumulative NEP and NCE than the other runs (Fig. 3 e,f).  These results show that
averaging the initial conditions is a good way to reduce cohort complexity.  The mapped patterns
(Fig. 4) show that large positive NEP differences between the CONDENSED and RESTART
runs occur in the upper Midwest and central California, which are dominated by cropland (Fig.
S2b).  This results from the reinitialization process in which the NPP of cropland starts out larger
than after accounting for transient conditions.  Forested areas in the Southeast are lower NEP in
the CONDENSED, which would be expected of more mature forests.  Differences in the rest of the
the country are minor.  The largest differences in NCE are the negative differences in the
Southeast corresponding to the NEP differences there.  The lower NEP in the CONDENSED run
is the result of larger heterotrophic respiration ($R_h$) more than offsetting slightly larger Net
Primary Productivity (NPP).  Since NEP is the difference between NPP and $R_h$, the net effect is a
negative bias in NEP (Fig. 5).

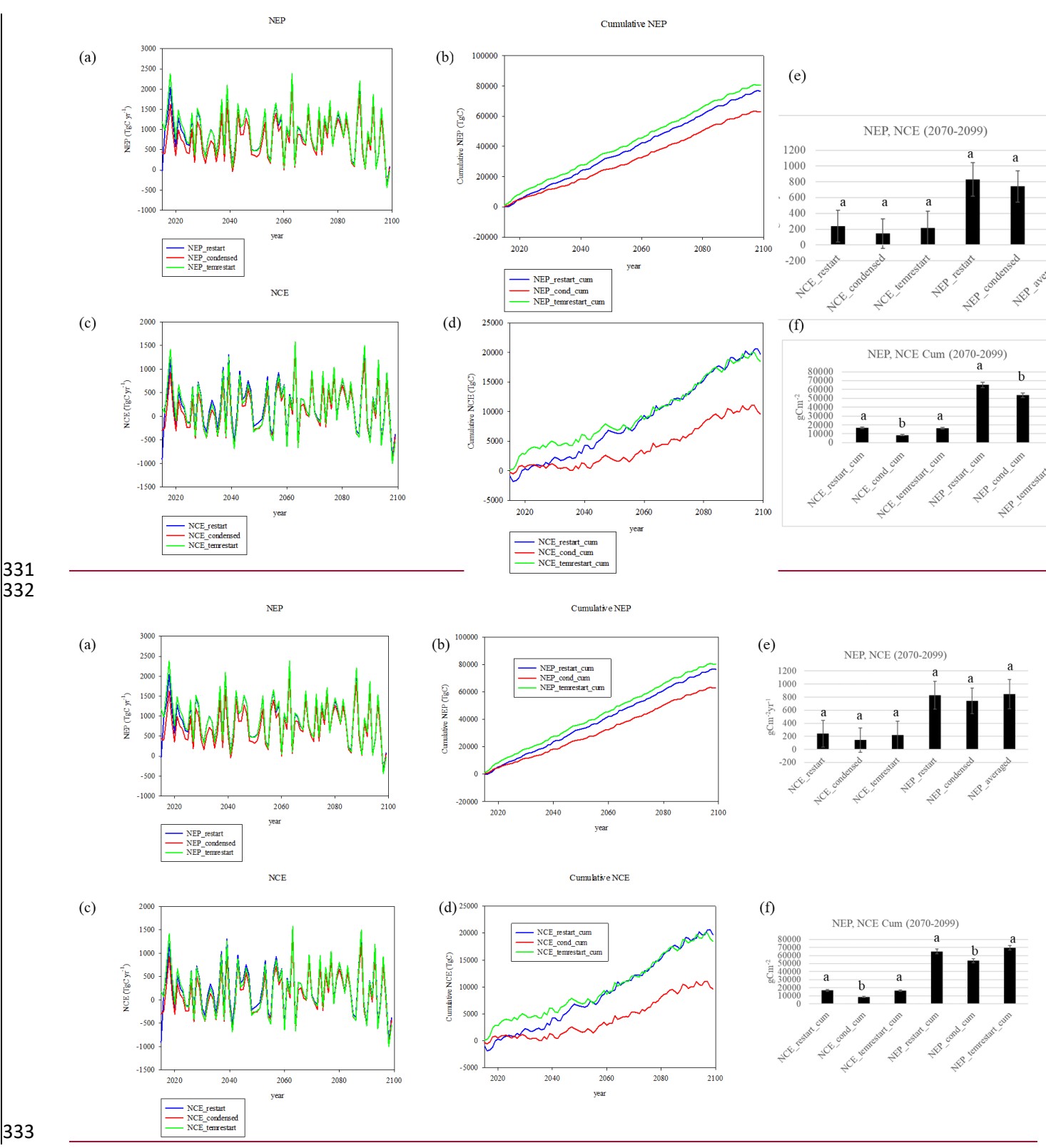

**Figure 3:** Comparison of NEP and NEC between the RESTART, CONDENSED, and
TEMRESTART runs, a) NEP, b) cumulative NEP, c) NCE, and d) cumulative NCE, e) NEP,
NCE comparison 2070-2099 means (error bars 1 standard deviation), f) cumulative NEP, NCE

comparison, 2070-2099 means (error bars 95% confidence interval).  ANOVA analysis for d and
e based on P<0.05.

(a) 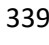

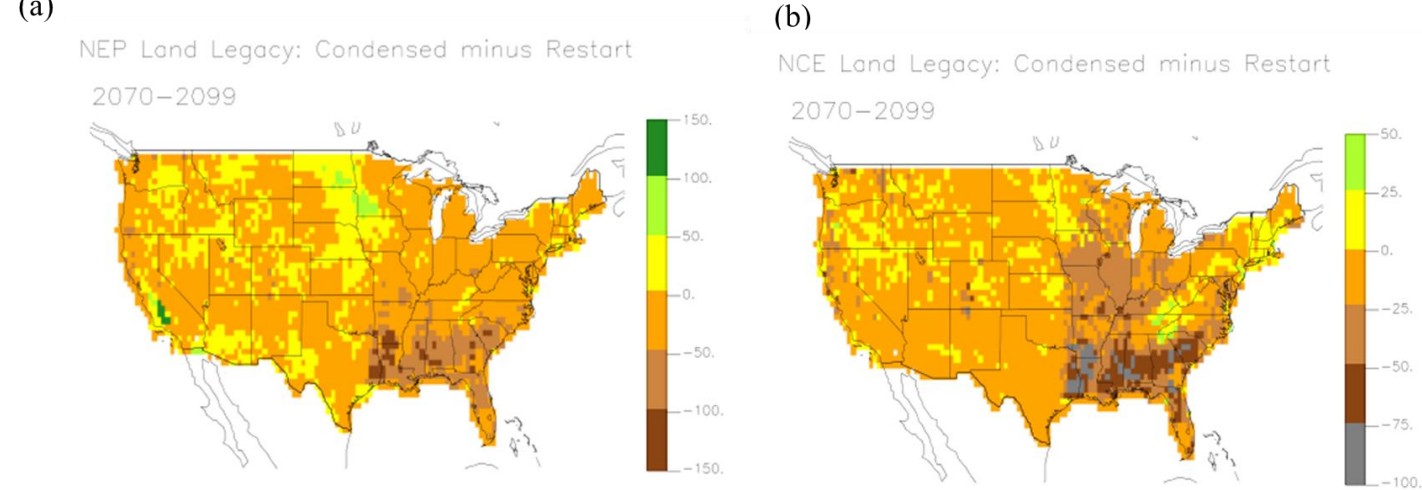

**Figure 4:** Mapped differences in NEP and NCE, illustrating effect of land legacy as difference
between the CONDENSED and RESTART runs, a) NEP (-164 to 198 gCm-2yr-1)), b) NCE (-
164 to 55 gCm-2yr-1).

(a) 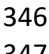

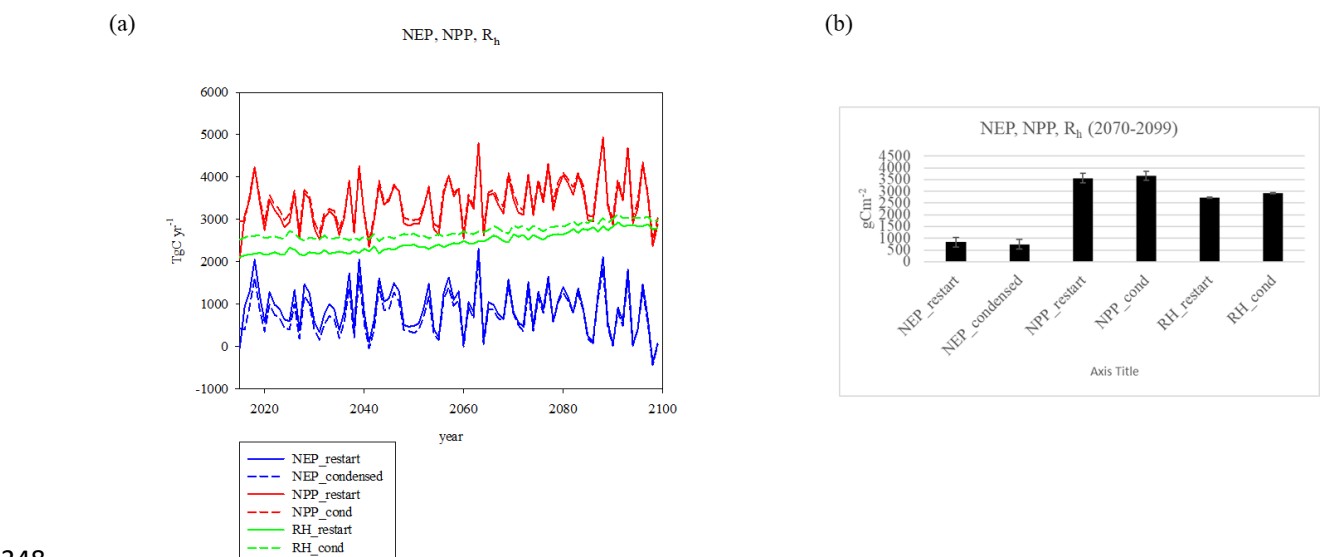


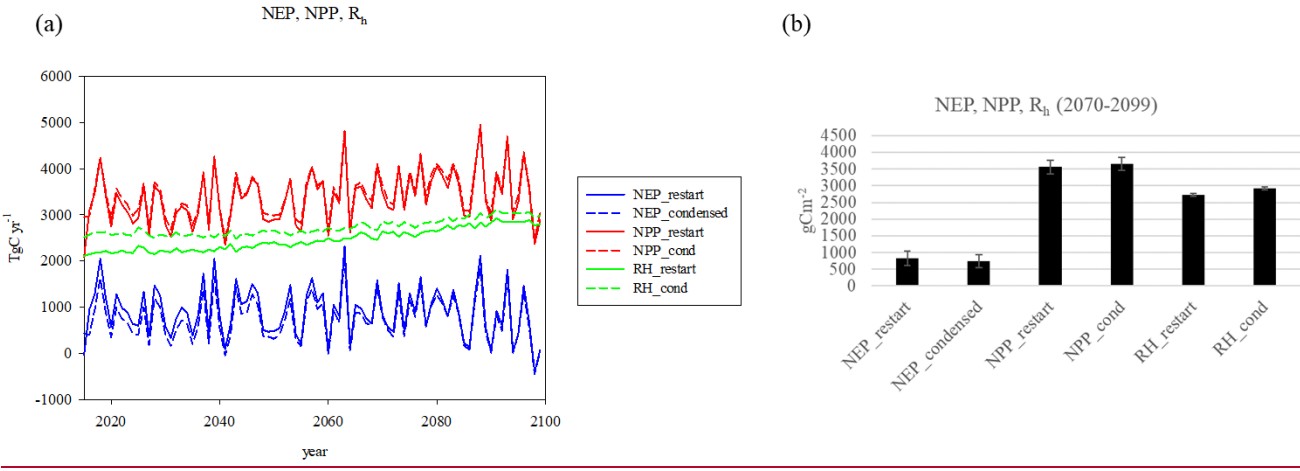

**Figure 5**: Carbon fluxes (NEP, NPP, Rh) for the RESTART and CONDENSED runs, , b) mean differences 2070-2099 (no significant differences for either of the three pairs).

While the more mature forests in CONDENSED would be expected to have lower NEP (Besnard et al., 2018; He et al., 2012), they would also have more biomass.  By the end of the century regrowing forests in the RESTART run will still be younger than those in CONDENSED run, and 85 years is not enough time to reach full equilibration in the model.  The CONDENSED vegetation carbon is 14% higher than the RESTART value by the year 2099, while the TEMRESTART is only 5% higher (Fig. 6).  The larger values in the CONDENSED run is due to the fact that the larger percentage of mature trees (since all trees are considered mature in the CONDENSED run) result in much more biomass.  Starting with averaged initial conditions lowers the vegetation carbon so that it is close to that of using the full cohorts.  The soil carbon is 31% higher in the CONDENSED run, while differences are minimal with the TEMRESTART run (Fig. 6).  Note that the absolute differences are larger with vegetation carbon, while the percent differences are more similar since the soil carbon has lower absolute values.  The mapped pattern of vegetation carbon differences between the CONDENSED and RESTART runs (Fig. 7a) shows that the large positive bias results almost entirely from the eastern half of the U.S., especially in the forested eastern portion, while the West exhibits smaller negative biases.  The soil carbon differences (Fig. 7b) are more scattered, with largest positive biases along the East coast and negative biases largest in the Southwest U.S. or Great Plains.

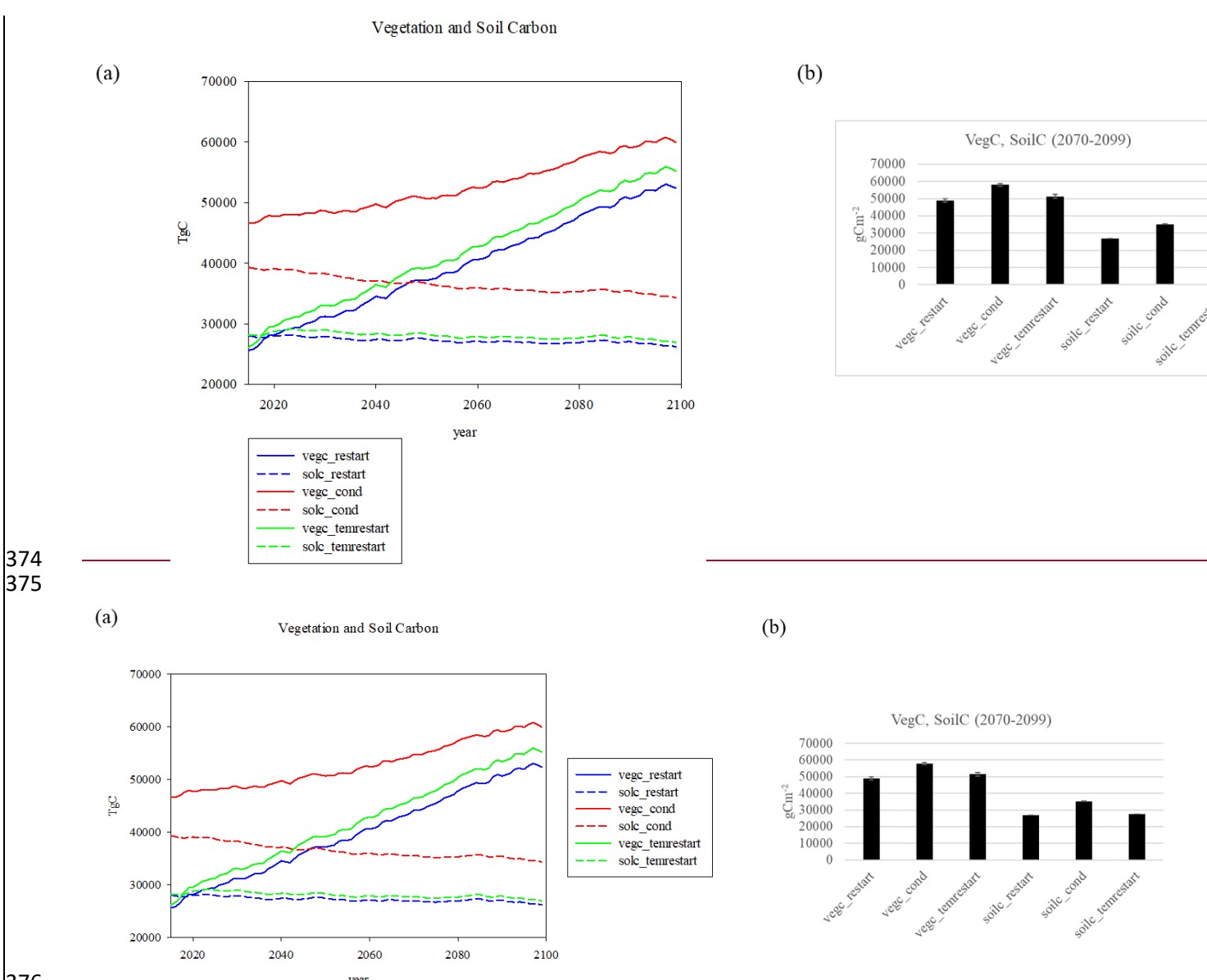

**Figure 6:** Vegetation and soil carbon in the RESTART, CONDENSED, and TEMRESTART experiments, b) mean differences 2070-2099 (all three vegetation carbon and soil carbon differ significantly from each other).

(a)                                                             (b)

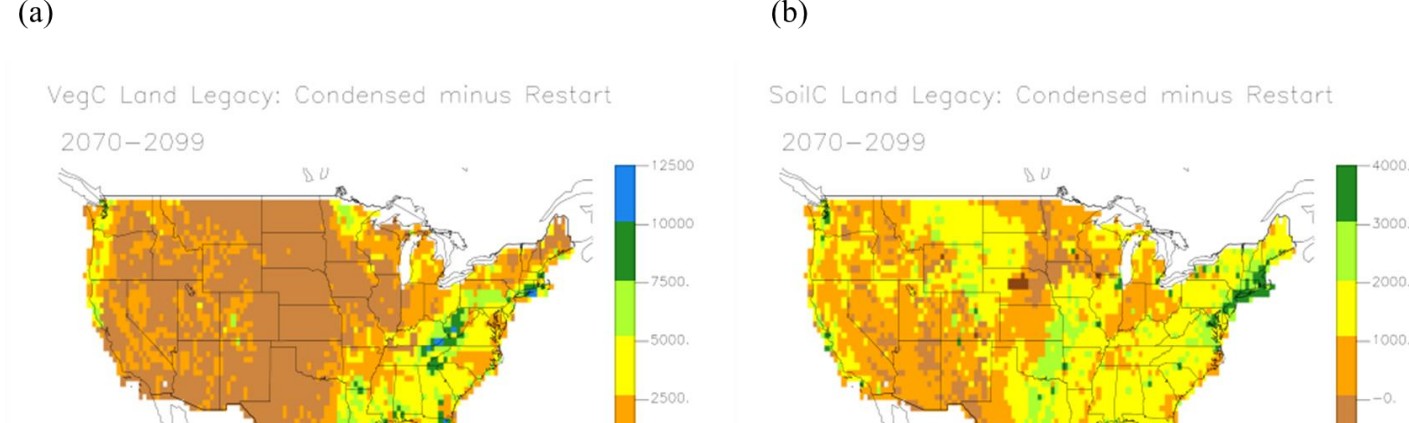

**Figure 7:** Mapped patterns in a) vegetation (-14350 to 13146 gCm-2) and b) soil carbon (-2489
to 9339 gCm-2) as differences between the CONDENSED and RESTART experiments.

The keys to these differences are the distribution of stand age in the forests and nutrients in the
soil during regrowth.  Forest stand age in 2014 at the start of the future runs (when there is no
further disturbance) shows that while the largest bin of tree area is mature trees (> 500 years
old), the next largest class is young trees less than 11 years old, with a majority of tree area less
than 71 year old, based on the disturbance history of the Hurtt et al. (2020) dataset (Fig. 8a).
However, the majority of mature forests are in the Western U.S.  Most of the forests in the
eastern U.S. are under 30 years old (Fig. 8b, S4).  The biomass is generally larger for the more
mature categories (Fig. 9a,b).  More mature trees are therefore more important to determining
biomass than an even relatively large portion of younger trees.  While biomass generally
increases with stand age, NEP peaks between 11-30 years (Fig. 9c,d).  When classifying
vegetation carbon by PFT (Fig. 10a), the CONDENSED run values are larger than the
RESTART values for boreal forest and temperature coniferous, deciduous, mixed, and
broadleaved evergreen forests, as well as savanna (which is a mixture of grassland and trees).
The NEP differences between CONDENSED and RESTART runs (Fig. 10b) shows NEP is
generally lower in the CONDENSED runs since each cohort has been reinitialized at the start,
but the interannual variability (IAV) is much larger than the differences.

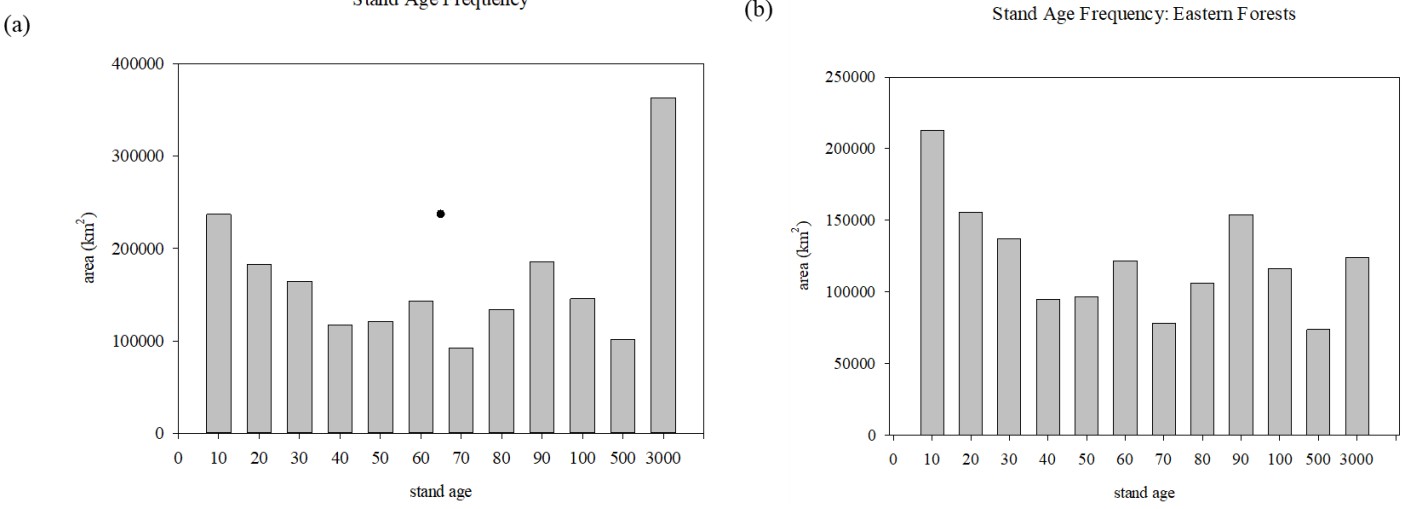

**Figure 8:** a) Stand-age frequency for U.S. and b) eastern forests. Bins represent 0-10, 11-20, 21-
30, 31-40, 41-50, 51-60, 61-70, 71-80, 81-90, 91-100, 101-500, > 500 years.

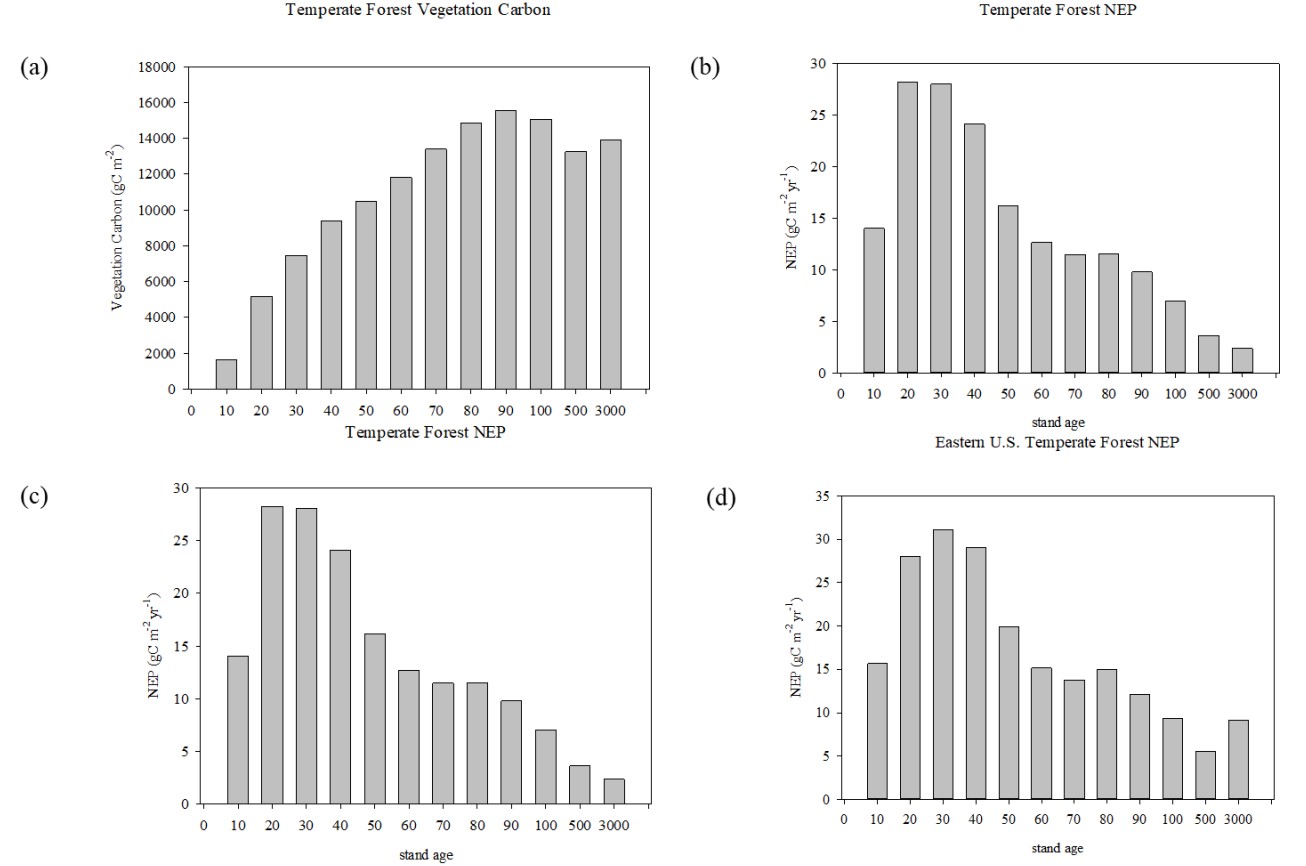


**Figure 9:** Vegetation carbon in the year 2014 for a) U.S. and b) eastern forests, and NEP in the year 2014 for c) U.S. and d) eastern forests. Most trees are not mature, but the mature trees contain the most biomass, so condensing the cohorts overestimate vegetation carbon.

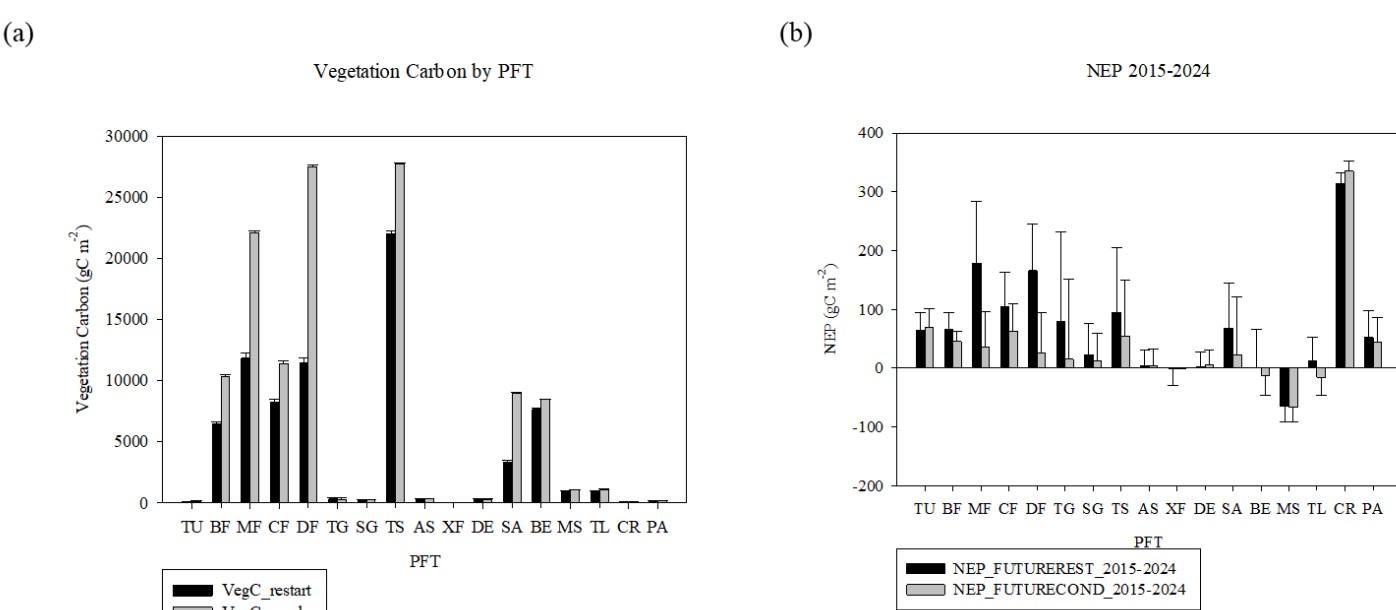

(a) Vegetation Carbon by PFT

(b) NEP 2015-2024

**Figure 10:** a) Vegetation carbon by plant functional type in 2015 for the RESTART and for the CONDENSED experiments, b) NEP by PFT averaged 2015-2024. PFTs are:TU = tundra, BF = boreal forest, MF = mixed temperate forest, CF = temperate coniferous forest, DF = temperate deciduous forest, TG = tall grasslands, SG = short grasslands, TS = tropical savanna, AS = arid shrublands, XF = xeric forests and woodlands, DE = deserts, SA = temperate savannas, BE = temperate broadleaved evergreen forests, MS = Mediterranean shrublands, TL = turflawn, PA = pasture, CR = crops. Error bars are 10-year interannual variability, computed as 95% confidence interval of year 2015-2024 for each PFT in each of the two runs.

The inorganic nitrogen in the soil is crucial for regrowth following disturbance. The dependence of available inorganic nitrogen following a disturbance on the final vegetation carbon by the year 2100 is generally a positive slope, but there is a lot of variability due to so many other factors affecting forest regrowth. Illustrated here (Fig. 11) is the amount available when disturbance occurs just before forest regrowth vs the final vegetation carbon for that cohort in the year 2100, since no further disturbance occurs in the future. Only values of inorganic nitrogen < 10000 mgN/m2 are shown, because larger values of available nitrogen are not limiting to forest growth. It is evident that Larger amounts of initial inorganic nitrogen generally lead to greater forest growth, as long as values are low enough to be limiting. although there is a wide range in the slope of that relationship. There are also many cohorts that have low growth regardless of initial nitrogen levels, so they are limited by other climate or environmental factors. This is only illustrated true for the more mesic forests of the eastern U.S. where moisture is less limiting. The final amount of available inorganic nitrogen in 2100 will be compensated by the fact that mature

forests provide more nutrients because of the greater litter but also use more nutrients due the
higher biomass.

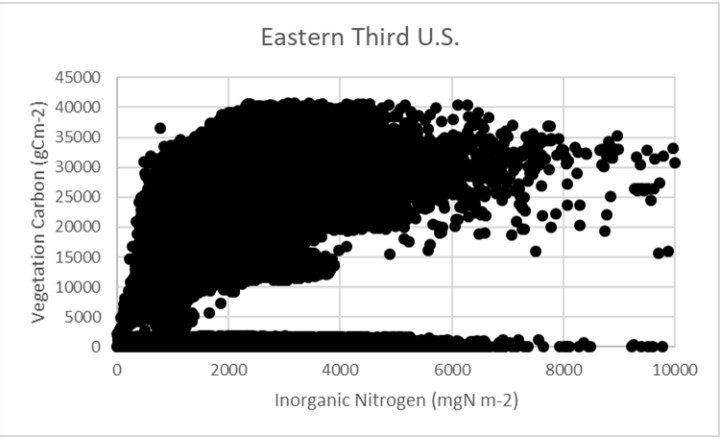

**Figure 11:** ~~Inorganic nitrogen available for plant uptake immediately following disturbance~~
~~before forest regrowth vs the final vegetation carbon by 2100.  Shown here are value so~~
~~inorganic nitrogen less than 10000 mgN/m2.~~
The soil moisture is based on a bucket model and accounts for the excess of precipitation over
evapotranspiration, with runoff resulting if the bucket (whose capacity equals the difference
between field capacity and wilting point) is overflowed. The soil moisture of the CONDENSED
run over the last 30 years is not statistically different from RESTART, while the TEMRESTART
~~is 1.8%~~though higher, is not significantly different during that time period (Fig. 1~~1~~2a, d).  The
evapotranspiration flux of the CONDENSED run is too low compared to RESTART while it is
too high in the TEMRESTART run, but the runoff fluxes are nearly identical between the three
runs (Fig. 1~~1~~2-b,c,e).

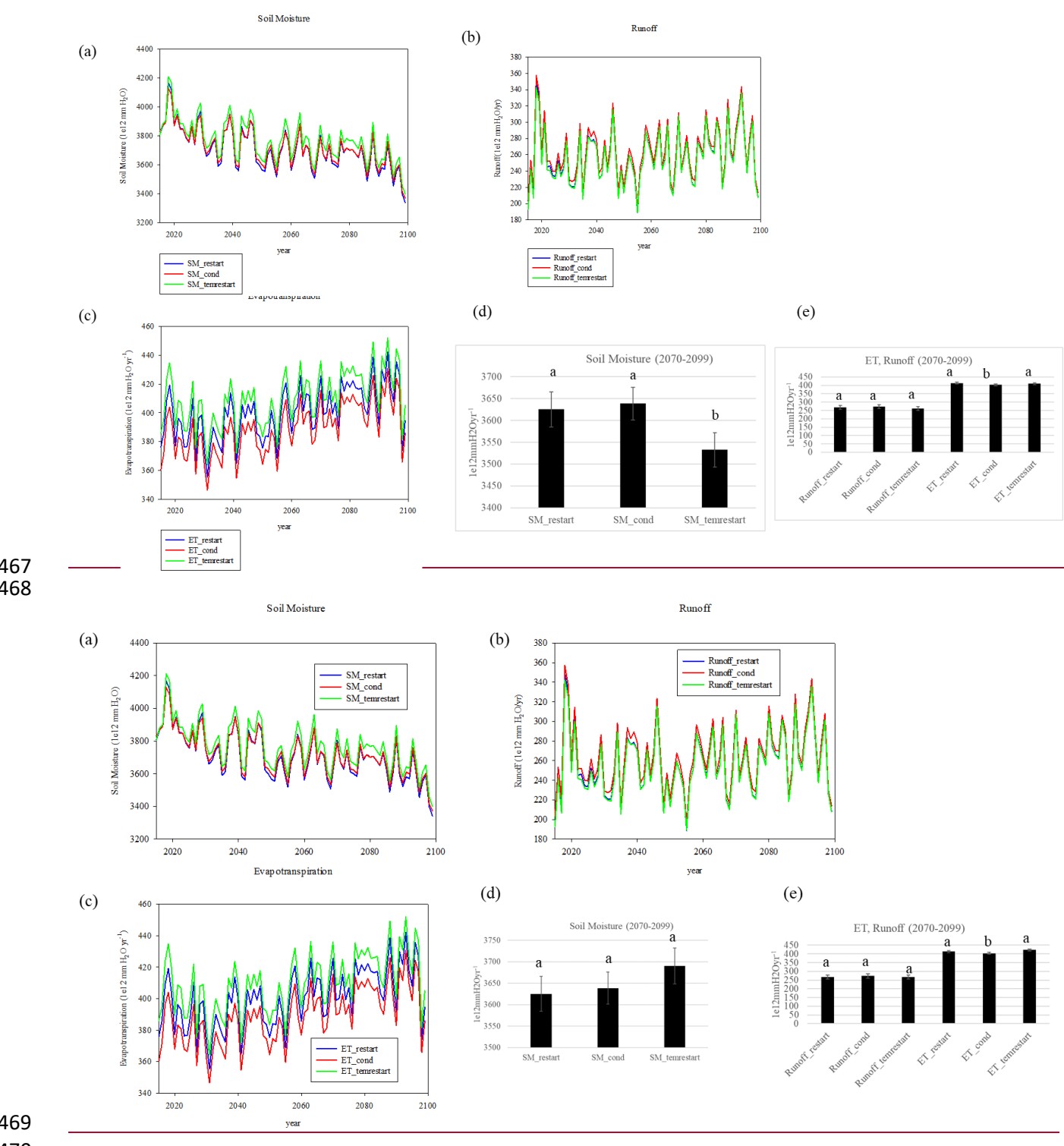

**Figure 112:** a) Soil moisture b) runoff, and c) evapotranspiration between the RESTART, CONDENSED, and TEMRESTART experiments, d) mean differences 2070-2099 for soil moisture, and e) mean difference 2070-2099 for runoff and evapotranspiration (letters based on ANOVA analysis with P<0.05).

**4 Discussion**

The measured stand age frequency in the U.S. is given in Pan et al. (2011) for different regions of the U.S. The eastern regions are dominated by younger trees, the Rocky Mountains by more mature trees as well as a peak in very young trees, and the West coast more younger and mid-age trees. Lu et al. (2015), using a similar LULCC dataset as used here based on Hurtt et al. (2011) land use transitions, specifically corrected that dataset to better represent the data from Pan et al. (2011). The resulting correction was younger forest stand ages in the eastern U.S. after 1850, with overall younger stand ages in the conterminous U.S. as a whole. In fact, the stand age distribution for the NE U.S. before the correction (Fig. S2 in (Lu et al., 2015) shows most forest older than 70 years, whereas the Pan et al. (2011) data show most forests are younger. The more recent land use dataset developed from Hurtt et al. (2020) actually shows a majority for forests in the eastern U.S. less than 70 years old (Fig. 8), but for the conterminous U.S. the frequency of mature forests is larger because of forests in the western U.S.

Total biomass increases with age, such that more mature trees have higher amounts of vegetation carbon (Chapin Iii et al., 2011; Pan et al., 2002), consistent with the results presented here (Fig. 9a,b). The slight decrease in biomass for some of the more mature stand age classes can represent the differences between geographic areas in which different classes dominate, as biomass for similar trees will be larger under more favorable climate conditions. For example, more mature trees in intermountain forests in the Western U.S. may be expected to have less biomass than less mature trees in the more mesic Eastern U.S. In the eastern U.S. the 101-500 year class, for example, the reduction in biomass is due to trees in the northeast (Fig. S5). Note that there is no explicit mortality modeled in TEM-Hydro, so biomass in mature forests is not decreasing because of increased mortality, which is another cause for reduced biomass in old stands (Xu et al., 2012). The mapped differences at the end of the $21^{st}$ century (Fig. 4, 7) represent the aging of all forests in the experiments, so the age distribution in the RESTART run would be shifted upward by 70 years, so all the forests will be in the upper age categories in both RESTART and CONDENSED runs. Positive biomass differences in the eastern U.S. (Fig. 7a) may represent the even more mature status of the forests in the CONDENSED runs in that region. Forests in the CONDENSED run would be expected to have lower NEP since they are more mature, which is generally true of forests, especially in the Southeast U.S. (Fig. 4a), but by the end of the century all the forests have matured more in the RESTART run as well, so differences are more muted with time.

NEP generally peaks between 20 and 30 years stand age, yet remains positive for hundreds of years (Luyssaert et al., 2008). The TEM-Hydro results from the HISTORICAL run show maximum NEP occurring between 11-30 years for temperate forests across the U.S. or up to 40 years in the eastern U.S. (Fig. 9 c,d), with NEP generally remaining positive except for very old trees when including the Western U.S. In fact for the conterminous U.S. as a whole, Lu et al. (2015) found that the Pan et al. (2011)-corrected data, with much younger stand age distribution, had a cumulative NCE of 323 TgC/yr from 2001-2005 vs 173 TgC/yr with the uncorrected data derived from Hurtt et al. (2011). The RESTART and TEMRESTART runs show continued carbon uptake in the future (Fig. 3), consistent with Krause et al. (2020)) who point out that regrowth, as well as climate change and elevated $CO_2$, will continue to promote carbon uptake

even in the absence of future land use change. Houghton et al. (2012) also explains that future carbon uptake is dominated by land legacy effects.

The interannual variability of fluxes, like NEP and NCE is very large (493-579 TgC/yr standard deviation, or over 63-75 gC/m2/yr for the three runs for NEP), so the differences between the experiments are all within the interannual variability. These values are consistent with other measured values. For a range of 24 eddy covariance sites, standard deviation of annual Net Ecosystem Exchange (NEE) ranged from about 20-280 gC/m2/yr, accounting for 50% of annual NEE (Niu et al., 2017). IAV from site-level FLUXNET sites mostly in North America and Europe ranged from 15 to 400 gC/m2/yr (with a mean of 130 gC/m2/yr), with lower values in more northern sites, and a lower range of values from global upscaling and inversion models (Marcolla et al., 2017). Climate drivers, particularly temperature and moisture, are considered the primary drivers for this large IAV (Piao et al., 2020). In any case, differences between the experiments in this study (Fig. 3) are all much smaller than the IAV, but the different experiments are well correlated, so the differences represent a shift of the entire time series, rather than a change in IAV.

The effect of nutrient loading on abandoned land, such as fertilization on abandoned cropland, can increase the final growth of the forest, but final growth rates are dependent upon many other environmental factors as well, which is why the relationship does not hold true everywhere, and above a certain level of nutrient availability, the system is not nitrogen limited, so it does not matter at all (Fig. 11). Other studies have confirmed that increased nutrient availability, in the form of lower C:N and C:P or high P, promotes radial stem growth (Mausolf et al., 2018) or tree ring width (Von Oheimb et al., 2014), which is consistent with the biomass results from this study. The greater nutrient availability, by directly increasing GPP, would also result in more litter and therefore more litter decomposition and higher rates of net nitrogen mineralization, also consistent with Von Ohemib (2014). However there is also a legacy effect of reduced resiliency to drought, having to do with changes in soil structure, which would not occur in the model development here.

Most other terrestrial ecosystem models do not include the effect of forest demography. The Dynamic Global Vegetation Models (DGVM) included in Trends in Net Land–Atmosphere Exchange (TRENDY-v2) (Li et al., 2017) mostly include annual changes in PFTs to represent LULCC. They include the conversion and product fluxes resulting from these changes, and often include the effects of mortality and regrowth within existing grids, but do not incorporate the effects of forest regrowth due to LULCC. Two of the models (VISIT and JSBACH) (Kato et al., 2013; Reick et al., 2013) include elaborate methods of applying the LULCC transition matrices to ensure the correct redistribution of PFTs and correct carbon fluxes. Shevliokova et al. (2009) does use a tiling approach to consider forest stand age and reduce the large number of cohorts used here. The HISTCOND run was designed specifically to explore the effects of forest demography by trying to emulate the effect of just redistributing annual land-use fractions, without including the effect of forest demography or keeping track of soil nutrients. As seen in the results, it does substantially overestimate the carbon stocks and underestimate the NEP compared to the run that includes the full effects of forest demography.

Restarting from averaged initial conditions more closely approximates the full cohort approach
with a large computation advantage by avoiding the need for reinitializing and enabling the use
of condensed cohorts, but with the corrected initial conditions.  In the fluxes (Fig. 3), cumulative
NEP of TEMRESTART is higher than the RESTART run, but cumulative NCE of the
TEMRESTART is nearly the same as the RESTART run in the latter half of the century.  The
vegetation carbon of TEMRESTART diverges slightly from RESTART, while the soil carbon
barely diverges at all (Fig. 6).
To address the issue of discontinuity between using clouds as input for the historical period
(1750-2014) and net irradiance for the future (2015-2099), an additional FUTURE run was
implements to use clouds for the future period as well. The reason for using clouds historically is
because net irradiance is not available from CRU4.04 dataset.  The model, and actual
ecosystems, are affected more directly by net irradiance than clouds.  The model code is
designed to convert clouds to net irradiance if net irradiance is unavailable (Raich et al., 1991)
which means there can be considerable error in the net irradiance values calculated from cloud
data.  Therefore it is most accurate to correct the historical cloud data to the bias-corrected
MACA net irradiance, which is what was done in this study.  The additional run involved using
total cloud fraction output directly from the same r6i1p1 NCAR CCSM4 RCP8.5 simulation.
Note that since these data are not available from MACA, they were bias corrected and
downscaled to the corrected cloud data using the period 2006-2014 and a similar method as used
to bias correct and downscale the MPI model output to CRU.  The results are all statistically
insignificant differences in NEP, NCE, cumulative NEP, cumulative NCP, vegetation carbon,
and soil carbon.
Water variables depend upon precipitation (which is similar between the runs, but can be rain or
snowmelt) and evapotranspiration, which ultimately depends upon environmental conditions (i.e.
solar radiation, vapor pressure deficit), stomatal conductance, and soil texture (Felzer et al.,
2011; Shuttleworth and Wallace, 1985).  The CONDENSED run exhibits a lower ~~bias in~~
evapotranspiration than RESTART, which is primarily due to low values in pasture grids (Fig.
11~~2~~).  Pasture in the CONDENSED run has higher leaf area index (LAI) then in the RESTART
run, due to reinitializing from equilibrium conditions, and that reduced the net irradiance, which
limits the amount of soil evaporation.  The effect of LAI on soil evaporation in the Shuttleworth
Wallace or Penmon Monteith approaches takes the form of an exponential decay, resulting in a
much sharper dropoff in evaporation with smallerl changes in low LAI than large LAI, which is
why the effect is predominant in low height vegetation like pastures.  The soil moisture is
slightly too large in the TEMRESTART run even though it starts off at the correct value, which
also results in a larger evapotranspiration rate, though neither significantly different from
RESTART by the end of the century.  The larger biases in the evapotranspiration flux do not lead
to larger biases in the soil moisture stock.   While evapotranspiration depends upon vapor
pressure deficit, net irradiance, stomatal conductance and surface roughness, and its value affects
the soil moisture, the amount of soil moisture also affects the amount of water available for
evapotranspiration.  Increasing vegetation cover has competing effects of reducing soil moisture
by shading the ground and increasing evapotranspiration, yet the relative effect of the two
depends upon range of the LAI change.
**5 Conclusions**

This study explores the role of past land use and land cover legacy on the future carbon and
water dynamics of terrestrial ecosystems in the conterminous U.S. While most models
simulating the future start with current LULCC by reinitializing initial conditions, the actual
value of the initial conditions should be different because ecosystems are not in a state of
equilibrium, but are changing due to past disturbances and climate change. This study
determines whether it is nevertheless possible to use a single realization for each PFT if the
initial conditions are set correctly based on a past run that includes land use and land cover
legacy effects.

NEP, a measure of carbon sequestration, is too low compared to using all the cohorts when
reinitializing initial conditions because the assumption of mature forests rebalances the NEP to
become more neutral through enhanced heterotrophic decomposition. There are some offsetting
geographic differences across the U.S. when accounting for all ecosystems. The NCE
differences are somewhat reduced, however, due to continued product decomposition in runs that
account for transient changes to LULC in the past. Cumulatively, condensed cohorts have a
negative bias in both NEP and NCE, which becomes a positive bias in the case of NEP and is
eliminated in the case of NCE by the end of the century when initializing correctly
(TEMRESTART). This is evident in the larger values in the biomasses (vegetation and soil
carbon) relative to RESTART, which are too large for the CONDENSED cohorts but greatly
improved with TEMRESTART. When PFTs are condensed into single cohorts, the forests are
all assumed to be mature forests, which leads to an overestimate of the biomass. The NEP of
mature forests is generally less than that of younger forests, though the actual biases between the
CONDENSED and RESTART runs but the end of the century are more muted as the forests
have had a chance to mature more in both. Correcting for initial conditions reduces the bias in
vegetation carbon and eliminates the bias in soil carbon. Starting with the correct initial
conditions do not have a large impact on the water variables, as they are more dependent on
environmental factors, though the vegetation cover does have some minor effects.

In addition to forest stand age, the initial nutrient loading of the soil is also an important factor
for future forest regrowth. With low levels of nitrogen, higher starting values often lead to a
larger overall biomass as the forest develops, though there are other environmental factors (e.g.
climate) that are important. Past agricultural use could deplete the soil of nutrients if cropland
was abandoned at a time period before chemical fertilization was frequently used (i.e. before the
1950s), or could enhance the soil nutrients if abandoned from heavily fertilized soil. These
effects will be accounted for if the correct initial soil conditions are determined.

This study illustrates the importance of accounting for the correct forest stand age and initial soil
nutrient conditions in order to model the future carbon sink. Although starting model runs in the
1700s or earlier is computationally expensive, it is possible to average values from such a run for
each PFT to allow a run to start in the present with correct initial condition and achieve a result
more consistent with a detailed representation of land-use cohorts. While this research assumed
constant LULC for the future, the next step is to use the corrected initial conditions as a basis for
future LULCC. A similar approach can be used to start land use transitions at any particular year
based on the complete history of land use transitions from 850 A.D. to serve as starting
conditions for one of the SSP scenarios.  Modeling groups need to consider this effect of past
LULC legacy to accurately estimate future carbon biomass and fluxes.
**Acknowledgments**

I would like to thank Yuning Shi and Dork Sahagian for reviewing this document, as well as my
graduate students (Christopher Andrade, Dannielle Waugh, and Jared Kodero) for patiently
discussing this research at our weekly research meetings.

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
