# Peer review of "Effect of land use legacy on the future carbon sink for the conterminous U.S. Benjamin S.Felzer Earth and Environmental Sciences Lehigh University 1 W. Packer Ave. Bethlehem, PA 1807 bsf208@lehigh.edu"

_Biogeosciences, 2022_

## Referee Comment (RC1)

MS Title: Effect of land use legacy on the future carbon sink 1 for the conterminous U.S.

Author: Benjamin S.Felzer

**General comment**

The manuscript entitled "Effect of land use legacy on the future carbon sink for the conterminous U.S." by Benjamin S.Felzer assesses the effects of land use and land cover change (LULCC) from 1750 to 2014 using the Terrestrial Ecosystems Model (TEM) for the conterminous U.S. Comparing model outputs from four experiments, the author finds that LULCC legacy has a considerable impact on carbon pools and fluxes. While the topic is of general interest, I don't follow the experimental design of the study. My primary concern is that the author combines meterological forcing data from one climate model (MPI-ESM-P 1750-1900) with quasi-observed data (CRU, 1901-2014) and climate projections from another model (CCSM4, 2015-2099) in one continuous simulation. Each data set comes with their own sets of biases. This lack of consistency will manifest as a forcing, causing changes in carbon fluxes. So the final signal is a combination of changes in environmental conditions, such as climate change, as well as the transition between inconsistent data sets. Also, one of the experiments (CONDENSED) has been re-equilibrated to the environmental conditions of 2015-2045. It remains unclear to me why the author chooses to equilibrate the model with projected climate conditions when trying to isolate the role of LULCC. I recommend that the author either adjusts the methods, or provides additional explanations that justify his approach. I recommend that the manuscript may be considered for publication in Biogeosciences after major revision.

**Detailed Comments**

L19 It is common practice to account for LULCC that started during the pre-industrial period. For instance, the TRENDY model ensemble that informs the annual publication of the global carbon budget accounts for LULCC starting in the year 1701.

(see https://blogs.exeter.ac.uk/trendy/protocol/)

L41 To my understanding you don't compare model output against observation-based reference data. I would therefore not write that "carbon stocks are overestimated". Instead, I would either describe how carbon stocks differ among experiments or expand the analysis by comparing results against observations.

L55 Replace "address" with "addresses".

L80 Would a carbon sink related to regrowth not be larger if disturbance rates reduce, rather than "continue"?

L80 Spell out the FIA acronym.

L149 This section describes the different experiments (historical, restart, condensed, and tem-restart) with respect to their initial values and whether they are based on the full or condensed version of the cohort. Please add how you treat LULCC, atmospheric CO2 concentrations,

nitrogen deposition and nitrogen fertilization when describing each experiment, and include this information in the table.

L156 Please motivate why you condense the full cohorts to 1 cohort/PFT.

L166 You write that "the difference between the RESTART and CONDENSED runs shows the effect of including land legacy on future carbon dynamics". The difference between both runs is that the CONDENSED run re-equilibrates using climate data from 2016-2045 (L210). If the CONDENSED run has been re-equilibrated, then it is also in equilibrium with respect to the meteorological forcing, CO2, and N deposition + fertilization that correspond to 2016-2045. How do you separate the impact of land legacy from these other factors then? Please clarify or adjust your method.

L183 You Combine climate model data from one model, (MPI-ESM-P 1750-1900) with quasi-observed data (CRU, 1901-2014) and climate projections from another model (CCSM4, 2015-2099) in one continuous simulation. The more conventional approach is to conduct simulations that are either based on quasi-observed data or on data from one climate model. As for the quasi-observed climate data you could have used an early chunk of the historical data (e.g. 1901-1920) and spun up the model by iterating this climate data for whatever period it takes to equilibrate your model. This would have also freed you from the need of bias correction and downscaling MPI-ESM-P. The problem with your approach is that you combine data that come with their own set of biases and thereby mix the forcing from environmental factors with differences between these data sets. Please provide an explanation that justifies your experimental setup or adjust your method.

L197 It would be more convincing if you had used radiation rather than cloud data for the historical period as well. The change in your method from one period to the next may create an unnecessary artificial forcing, which then mixes with the impact of climate change. Please justify your approach or adjust your method. t

L210 You write that the CONDENSED run is first equilibrated based on repeated use of the 2016-2045 climate. It is not clear to me why you use projected future climate conditions to equilibrate your model. Please explain or adjust your method.

L211 Please define NCE before using the acronym.

L218 Please mention in the text what experiments you are referring to.

L238 You write that "reinitializing each grid is based on the assumption of NEP as close to zero". Should it not be Net Biome Productivity (NBP) rather than NEP that should be close to zero, as NBP also includes fluxes associated with disturbances, such as wildfires?

L275 Please explain why you expect that more mature forested in CONDENSED would have lower NEP. Also, please mention what period you are referring to. Finally, replace "mature forested" with "mature forests", if this is what you mean.

L278 The CONDENSED run has been re-equilibrated to the environmental conditions of 2015-2045. Is that not the main reason why vegetation carbon is 16% larger compared to the RESTART simulation? Also, I don't recommend using the term bias here if you are not comparing against observation-based reference data (here and elsewhere).

L280 Explain exactly what you mean by "fixes most of the problem".

L364 A difference of 1.0% or 1.8% does not seem very large and may not even be statistically significant.

L391 I believe it is Chapin "III" et al. Also, this statement seems a little vague to me. Please explain what you mean.

L411 Please mention what simulation you are referring to.

L433 One reason why the modelled NEE IAV is smaller than observed may be related to the fact that your model does not represent mortality. That may be worth mentioning here as well.

L481 I assume that "models of the future" refer to models that that project future changes in vegetation dynamics? Please rephrase.

L519 To decide whether a simulation is more realistic, you would need to evaluate your model output against some kind of observation-based reference data.

**Tables**

Table 1: Add information on how CO2, climate, and LULCC are treated in each experiment.

**Figures**

Figure01 The time axis covers the period 1750-2014. Why does the caption say that the curves corresponds to the HISTORICAL and RESTART run, if the RESTART run starts in 2015?

S1b Please make larger to enhance readability.

---

## Author Response (AR2)

Dear Dr. Felzer,

Thanks again for your hard work on this and I am happy to accept the article for publication. You have done a really thorough job through the review process and again that should be commended; so thanks!

I wish you luck with finalising the publication and your future research. Best regards, David

Non-public comments to the Author: I just have a couple of technical request with regards to the new figures:

Figure 2: the legend for panel (a) overlaps the axis title. My recommendation would be to move the figure legends over white-space within the actual confines of the graph. This will save space in the actual publication.

**Done. I also did this figure Figures 1 and 3.**

Figure 3: The presentation of Figure 3 is still slightly off and could be improved. Can you please remove the outline from panels (e) and (f), and also align these figures with panels (b) and (d), respectively.

**Done.**

Figure 5 and Figure 6: Please remove the outline from panel (b), update the axis title (currently: "axis title"), and I believe you could increase the height of the vertical axis. I would also recommend moving the Figure legend for Panel (a) to the space between the two panels (shifting panel (a) a little to the left).

**Done.**

Figure 10: Align the heights of the vertical axes in the two panels so they are the same height.

**Done.**

Figure 11: Is this figure completely necessary? It could potentially be included with your published data set. If you believe it is, please remove the figure outline, reduce the size of the data markers (by a lot), and fix the axis titles so the superscript values are actually in superscript.

Agreed. I removed the figure, but also had to revise the text accordingly (as well as relabel Figure 12 as Figure 11).

The inorganic nitrogen in the soil is crucial for regrowth following disturbance. The dependence of available inorganic nitrogen following a disturbance on the final vegetation carbon by the year 2100 is generally a positive slope, but there is a lot of variability due to so many other factors affecting forest regrowth. Larger amounts of initial inorganic nitrogen generally lead to greater forest growth, as long as values are low enough to be limiting. There are also many cohorts that have low growth regardless of initial nitrogen levels, so they are limited by other climate or environmental factors. This is only true for the more mesic forests of the eastern U.S. where moisture is less limiting.

The final amount of available inorganic nitrogen in 2100 will be compensated by the fact that mature forests provide more nutrients because of the greater litter but also use more nutrients due the higher biomass.

Figure 12: Align the panels, legends into white-space of graphs, remove outlines.

**Done.**

I also noticed that Figure 11d had incorrect values for the TEMRESTART run when comparing to the time series (Figure 11a). I had incorrect values for the TEMRESTART run for all the 30 year charts, so I corrected all of them, but the only on that is visibly different is Figure 11d. I also redid the SPSS ANOVA analysis to ensure the correct a and b values. Interestingly the soil moisture values are now not significantly different at all between the three cases during the final 30 years. Changes in test are in results paragraph on water:

The soil moisture is based on a bucket model and accounts for the excess of precipitation over evapotranspiration, with runoff resulting if the bucket (whose capacity equals the difference between field capacity and wilting point) is overflowed. The soil moisture of the CONDENSED run over the last 30 years is not statistically different from RESTART, while the TEMRESTART though higher, is not significantly different during that time period (Fig. 11a, d). The evapotranspiration flux of the CONDENSED run is too low compared to RESTART while it is too high in the TEMRESTART run, but the runoff fluxes are nearly identical between the three runs (Fig. 11b,c,e).

And discussion paragraph on water:

Water variables depend upon precipitation (which is similar between the runs, but can be rain or snowmelt) and evapotranspiration, which ultimately depends upon environmental conditions (i.e. solar radiation, vapor pressure deficit), stomatal conductance, and soil texture (Felzer et al., 2011; Shuttleworth and Wallace, 1985). The CONDENSED run exhibits a lower evapotranspiration than RESTART, which is primarily due to low values in pasture grids (Fig. 11). Pasture in the CONDENSED run has higher leaf area index (LAI) then in the RESTART run, due to reinitializing from equilibrium conditions, and that reduced the net irradiance, which limits the amount of soil evaporation. The effect of LAI on soil evaporation in the Shuttleworth Wallace or Penmon Monteith approaches takes the form of an exponential decay, resulting in a much sharper dropoff in evaporation with smallerl changes in low LAI than large LAI, which is why the effect is predominant in low height vegetation like pastures. The soil moisture is slightly too large in the TEMRESTART run even though it starts off at the correct value, which also results in a larger evapotranspiration rate, though neither significantly different from RESTART by the end of the century. The larger biases in the evapotranspiration flux do not lead to larger biases in the soil moisture stock. While evapotranspiration depends upon vapor pressure deficit, net irradiance, stomatal conductance and surface roughness, and its value affects the soil moisture, the amount of soil moisture also affects the amount of water available for evapotranspiration. Increasing vegetation cover has competing effects of reducing soil moisture by shading the ground and increasing evapotranspiration, yet the relative effect of the two depends upon range of the LAI change.

Finally, I have neatened up the figures throughout, in terms of sizing and alignment, which is why there is a new version of the SI.

**Dear Dr. Felzer,**

I would firstly like to commend you on your very thorough and considerate responses and effort in generating the necessary data on new modelled simulations to support your responses. These were quite enjoyable discourses to follow.

I would request just a few minor additions on top of the work that you have already done, which are in response to review comments as follows:

**Reviewer 1 L197.**

I would request that you add a simulation, likely to the SI would be sufficient, that includes only cloud data throughout the entire simulation and compare this to the run that switches between historical cloud data and irradiation when the irradiation data set becomes available. Then briefly discuss the findings, which should shed light on any data discontinuity concerns related to this issue.

Ok, I have done the requested run. In thinking about it, I realized that the MACA net irradiance is bias-corrected, so that was the justification for correcting the clouds to that (i.e. to ensure that the net irradiance from the historical period produced from the clouds matched it). The TEM routine actually converts clouds to net irradiance, which is what the model (and ecosystems) actually care about. So, I trust net irradiance data more than cloud data for forcing the model. I then realized that cloud data are not available from MACA. So, I went directly to the CMIP5 databased to download total cloud fraction from the NCAR CCSM4 RCP8.5 model of the correct ensemble (which was not easy to get as it was not available on the LLNL data portal where I usually get my model data, but I did find it on the CEDA portal). I then bias-corrected and downscaled that as described below to the corrected CRU clouds, and did the rerun for the FUTURE, which provides a continuous run with clouds. The results are barely different. In fact, showing a plot of the two (original and new FUTURE) runs is pointless because the curves are imperceptibly different. Therefore I don't think it makes sense to include anything new in the SI, but instead to add the paragraph below to the Discussion.

To address the issue of discontinuity between using clouds as input for the historical period (1750-2014) and net irradiance for the future (2015-2099), an additional FUTURE run

was implements to use clouds for the future period as well. The reason for using clouds historically is because net irradiance is not available from CRU4.04 dataset. The model, and actual ecosystems, are affected more directly by net irradiance than clouds. The model code is designed to convert clouds to net irradiance if net irradiance is unavailable (Raich et al., 1991) which means there can be considerable error in the net irradiance values calculated from cloud data. Therefore it is most accurate to correct the historical cloud data to the bias-corrected MACA net irradiance, which is what was done in this study. The additional run involved using total cloud fraction output directly from the same r6i1p1 NCAR CCSM4 RCP8.5 simulation. Note that since these data are not available from MACA, they were bias corrected and downscaled to the corrected cloud data using the period 2006-2014 and a similar method as used to bias correct and downscale the MPI model output to CRU. The results are all statistically insignificant differences in NEP, NCE, cumulative NEP, cumulative NCP, vegetation carbon, and soil carbon. (lines 572-586)

Raich, J. W., Rastetter, E. B., Melillo, J. M., Kicklighter, D. W., Steudler, P. A., Peterson, B. J., Grace, A. L., Moore Iii, B., and Vorosmarty, C. J.: Potential net primary productivity in South America: application of a global model, Ecological Applications, 1, 399-429, 1991.

**Reviewer 2 Major Concern (1):**

Please add a statement on the HISTCOND and lack of forest demography within this run. In your response to reviewer 2 point 1, you did not indicate that there was any change to the text regarding this additional run (which I believe is a very good inclusion). I think it is important to comment on this.

- From former review comments, added sentence in red (top paragraph) and black (bottom paragraph):
- This is an interesting point. After reviewing the full suite of TRENDY models, which serves as a nice intercomparison of leading edge DGVMs, I add that none of them actually include forest demography, though they do include changes in fractional land cover based on the LUC dataset, as well as conversion and product fluxes. I mention that Shiavlokova et al. (2009) does include forest demography using a tiling approach. In response to this comment, I have added a new experiment, HISTCOND, in which I condense all the cohorts in a transient historical run so that it is somewhat analogous to what many of the other models are doing. This new run does not include forest demography so that each year the fractional area of each condensed PFT is adjusted but no new cohorts are created to track soil carbon or forest stand age. This (and another new run referenced above as HISTCONST) are now in Figure 2, the Methods, Results, and Discussion. The key paragraph addressing this point in the Discussion is (lines 535-547):

Most other terrestrial ecosystem models do not include the effect of forest demography. The Dynamic Global Vegetation Models (DGVM) included in Trends in Net Land–Atmosphere Exchange (TRENDY-v2) (Li et al., 2017) mostly include annual changes in PFTs to represent LULCC. They include the conversion and product fluxes resulting from these changes, and often include the effects of mortality and regrowth within existing grids, but do not incorporate the effects of forest regrowth due to LULCC. Two of the models (VISIT and JSBACH) (Kato et al., 2013; Reick et al., 2013) include elaborate methods of applying the LULCC transition matrices to ensure the correct redistribution of PFTs and correct carbon fluxes. Shevliokova et al. (2009) does use a tiling approach to consider forest stand age and reduce the large number of cohorts used here. The HISTCOND run was designed specifically to explore the effects of forest demography by trying to emulate the effect of just redistributing annual land-use fractions without including the effect of forest demography or keeping track of soil nutrients. As seen in the results, it does substantially overestimate the carbon stocks and underestimate the NEP compared to the run that includes the full effects of forest demography. (lines 549-562)

**Reviewer 2 Major concern(3):**

I do note that I did request the removal of the exact text in the methods section published in a previous study of yours. This is a challenging situation and one I could not find a consensus on during discussion with the other associate editors. To move this forward, could you please summarize the requested information in a few sentences (with unique wording) and continue to use the existing reference to the details from your other studies?

Add this information to the Methods: This approach involves first using the LUH2 dataset to establish the fractional land cover type at the starting year of 1750. The primary and secondary vegetation are replaced with their potential vegetation values (as described in Raich et al., 1991), while other managed lands include croplands, pasturelands, and urban, with the multiple types of crops and pastures combined into single values for each, respectively. Disturbances (including timber harvest) involve the creation of new cohorts, with the corresponding area adjusted from the original cohort. Therefore, soil nutrients and forest stand age are tracked separately for each disturbance. The output are then area-weighted for each of the cohorts. Since this approach tracks each cohort separately, it is possible to end up with thousands of cohorts for a single grid cell by 2014. (lines 131-139)

Best regards,

Assuming the revised version of your manuscript reflects these and the other changes that you have indicated we can then proceed to acceptance and publication.

MS Title: Effect of land use legacy on the future carbon sink 1 for the conterminous U.S.

Author: Benjamin S.Felzer

**General comment**

The manuscript entitled "Effect of land use legacy on the future carbon sink for the conterminous U.S." by Benjamin S.Felzer assesses the effects of land use and land cover change (LULCC) from 1750 to 2014 using the Terrestrial Ecosystems Model (TEM) for the conterminous U.S. Comparing model outputs from four experiments, the author finds that LULCC legacy has a considerable impact on carbon pools and fluxes. While the topic is of general interest, I don't follow the experimental design of the study. My primary concern is that the author combines meterological forcing data from one climate model (MPI-ESM-P 1750-1900) with guasi-observed data (CRU, 1901-2014) and climate projections from another model (CCSM4, 2015-2099) in one continuous simulation. Each data set comes with their own sets of biases. This lack of consistency will manifest as a forcing, causing changes in carbon fluxes. So the final signal is a combination of changes in environmental conditions, such as climate change, as well as the transition between inconsistent data sets. Also, one of the experiments (CONDENSED) has been re-equilibrated to the environmental conditions of 2015-2045. It remains unclear to me why the author chooses to equilibrate the model with projected climate conditions when trying to isolate the role of LULCC. I recommend that the author either adjusts the methods, or provides additional explanations that justify his approach. I recommend that the manuscript may be considered for publication in Biogeosciences after major revision.

**Thank you for these comments.**

The issue of combining multiple climate datasets is an interesting one. I believe it is important to use actual climate data (i.e. CRU4.04) rather than modeled climate data for the historical period when it is available. Otherwise, modeled climates will not capture the correct interannual variability – i.e. warm/cold years, wet/dry years when they actually occur. The modeled "data" may have the correct trend, but will not be correct on a year-to-year basis. It is absolutely critical that the future climate model data be downscaled and bias-corrected, which is why I chose to use the MACA datasets. While MACA was bias corrected using GRIDMET, that dataset starts in 1979 so does not provide a long historical period of data. However the resulting time series are continuous so there are no large biases. The decision to use modeled data prior to 1901, from 1750 to coincide with the period of LULCC, was not taken lightly, as I originally used a more traditional approach of using the first 30 years of the historical climate repeatedly for the period 1750-1901. However, when I published the precursor to this paper (Felzer et al., 2018) a reviewer took me to task for that approach (earlier review comment "It is also difficult to understand why the model was run from 1700 since climate data were not available until 1901. I would think climate before 1901 has affected the carbon cycle in the US and that effect cannot be reproduced by using a 1901-1930 average climate that includes big fire years around 1910 and the beginning of the drought of the 1930s. The little ice age is not represented well by the 1901-1930 average climate"). For that reason, I responded by finding a millennial simulation, and chose the model with the highest resolution, the MPI-ESM-P (I also had to stitch together the millennial and historical periods

to go from 1750-1900). Importantly, I did my own downscaling and bias correction to ensure continuity of climate, which is evident in Figure S3. To thoroughly explain these issues to the reader, I added the following paragraph to the Experimental Design section (including 3 new references) – lines 239-250:

The decision to base climate prior to 1900, prior to the gridded historical data, was made to capture more realistic climate variations during the period from 1750 to 1900, such as the Little Ice Age (LIA), which lasted through the 19th century (Bradley and Jones, 1993; Mann, 2002). The temperature record from the MPI-ESM-P model does show signs of temperature climbing out of a cold peak after 1818 but remaining cool throughout the rest of the century (Figure S3), which is consistent with Northern Hemisphere proxy records (Mann et al., 2008). Since this study is for the conterminous U.S., it does not show as strong an LIA signal as would be expected from records in the North Atlantic. The decision to then use historical CRU4.04 climate rather than modeled climate from 1901-2014 is to more accurately capture the true interannual variability, which would be entirely lost by using output from a climate model. All three datasets have been downscaled and bias corrected to produce a seamless record of climate from 1750-2099.

The second point about equilibration period I agree warranted a model rerun of the CONDENSED experiment. The reviewer accurately points out that I should be using the prior 30 years to 2015, rather than the post 30 years, as the basis for equilibration. Therefore, I ran the dynamic equilibration from 1986-2015, and used that as the basis for the initial conditions of the CONDENSED experiment. This change is noted in the Experimental Design section (lines 249-250). The new results are now used in all the figures involving the CONDENSED run, and numerical values throughout the text changed, where necessary. This change did not significantly alter any of the results. Both old and new figures are present in the revised document, so the reviewer can confirm that the differences exist but are minor.

**Detailed Comments**

L19 It is common practice to account for LULCC that started during the pre-industrial period. For instance, the TRENDY model ensemble that informs the annual publication of the global carbon budget accounts for LULCC starting in the year 1701.

(see <a href="https://blogs.exeter.ac.uk/trendy/protocol/">https://blogs.exeter.ac.uk/trendy/protocol/</a>)

Thanks for the comment. I would say the industrial period begins with the start of the Industrial Revolution in the 1860s. However, in my earlier study (Felzer and Jiang 2018) I did start the runs in 1700. I actually made a conscious decision to change the starting period of this study to 1750 to be consistent with the IPCC AR6 report, which generally uses 1750 as their baseline. I first realized this when reviewing a paper in which the authors started in 1750 and referred to it as some standard IPCC baseline, which is why I researched the issue further and decided to start in 1750 rather than 1701.

L41 To my understanding you don't compare model output against observation-based reference data. I would therefore not write that "carbon stocks are overestimated".

Instead, I would either describe how carbon stocks differ among experiments or expand the analysis by comparing results against observations.

**Good point – fixed to read "The carbon stocks are larger than using all the cohorts if condensed cohorts ...". (line 41). I tried to correct this in other places as well.**

L55 Replace "address" with "addresses".

**Done**

L80 Would a carbon sink related to regrowth not be larger if disturbance rates reduce, rather than "continue"?

**I went back to the Pugh et al. (2019) paper, and their meaning is if disturbance rates continue at 1981-2010 levels, so I added "continue at historical levels" (line 81).**

L80 Spell out the FIA acronym.

**Done**

L149 This section describes the different experiments (historical, restart, condensed, and temrestart) with respect to their initial values and whether they are based on the full or condensed version of the cohort. Please add how you treat LULCC, atmospheric CO2 concentrations, nitrogen deposition and nitrogen fertilization when describing each experiment, and include this information in the table.

I replaced the table to include these new items, as well as ozone. Note that more detail is in the references or in the references of the associated text.

L156 Please motivate why you condense the full cohorts to 1 cohort/PFT.

Added the following sentence: The motivation for these two condensed-PFT runs is to reduce computational time by eliminating the need to run potentially thousands of land-use legacy cohorts for each grid when starting from present-day conditions (lines 184-187).

L166 You write that "the difference between the RESTART and CONDENSED runs shows the effect of including land legacy on future carbon dynamics". The difference between both runs is that the CONDENSED run re-equilibrates using climate data from 2016-2045 (L210). If the CONDENSED run has been re-equilibrated, then it is also in equilibrium with respect to the meteorological forcing, CO2, and N deposition + fertilization that correspond to 20162045. How do you separate the impact of land legacy from these other factors then? Please clarify or adjust your method.

This is an interesting point. However, the idea is to start a model in present-day conditions, you either need to provide it initial conditions, as in the TEMRESTART run, or reinitialize somehow. Two approaches to reinitialization are to use the 30-year average climate from, say, 1986-2014, or use those years for a dynamic equilibration, as I have done. Other conditions, such as CO2, N deposition, N fertilization, and ozone, will be the values pertinent to those 30 years, as will, in fact, the climate. So, I think that is the appropriate and only way forward. CO2, N deposition, N fertilization, and ozone changes all really ramped up in the latter part of the 20th century, and were nonfactors for most

of the prior period. I have added the sentences "Note that the RESTART run will also incorporate effects of changing climate, CO2, ozone, N deposition and fertilization, which cannot be captured in the CONDENSED run. (lines 188-190)" to acknowledge this point.

To separate out the effects of each would require factorial experiments running the model with only one factor, which is essentially what I did in the Felzer and Jiang (2018) study. However, to respond to this point further, I have done a new run (HISTCONST) in which land cover is held constant at 2014 values, so the difference between HISTORICAL and HISTCONST illustrates the effects of the other factors (note that since N fertilization is part of management, I do not list it along with N deposition when describing the environmental factors of change). I added this run into the Methods (lines 164-167) and Results (lines 285-292) sections, as well as included a new figure (Fig. 2) – that includes this run and another new run, HISTCOND, discussed below in response to the other reviewer.

L183 You Combine climate model data from one model, (MPI-ESM-P 1750-1900) with quasiobserved data (CRU, 1901-2014) and climate projections from another model (CCSM4, 2015-2099) in one continuous simulation. The more conventional approach is to conduct simulations that are either based on quasi-observed data or on data from one climate model. As for the quasi-observed climate data you could have used an early chunk of the historical data (e.g. 1901-1920) and spun up the model by iterating this climate data for whatever period it takes to equilibrate your model. This would have also freed you from the need of bias correction and downscaling MPI-ESM-P. The problem with your approach is that you combine data that come with their own set of biases and thereby mix the forcing from environmental factors with differences between these data sets. Please provide an explanation that justifies your experimental setup or adjust your method.

**Please see explanation provided above.**

L197 It would be more convincing if you had used radiation rather than cloud data for the historical period as well. The change in your method from one period to the next may create an unnecessary artificial forcing, which then mixes with the impact of climate change. Please justify your approach or adjust your method. T

This is a good point. However, net irradiance is not available in the CRU4.04 dataset, only clouds, which is why clouds were used. Added the following sentence: "The CRU4.04 data does not include irradiance, which is why it was necessary to use clouds for the historical period, but since net irradiance is more directly used by the model, that was chosen for the future period" (lines 230-232).

L210 You write that the CONDENSED run is first equilibrated based on repeated use of the 2016-2045 climate. It is not clear to me why you use projected future climate conditions to equilibrate your model. Please explain or adjust your method.

**True – as explained above, I did adjust the methods to use 1986-2014, and reran the CONDENSED run as a result.**

L211 Please define NCE before using the acronym.

Sorry, now defined, with added sentence, "NCE is the NEP plus carbon lost through landuse conversion or by decomposition of agricultural or timber harvest products." (lines 261-263)

L218 Please mention in the text what experiments you are referring to.

**I added (from HISTORICAL).**

L238 You write that "reinitializing each grid is based on the assumption of NEP as close to zero". Should it not be Net Biome Productivity (NBP) rather than NEP that should be close to zero, as NBP also includes fluxes associated with disturbances, such as wildfires?

The TEM approach is to equilibrate NEP. In TEM (from McGuire et al. 2001), the NCE is the flux term that includes fluxes associated with disturbances. The equilibration procedure does not include disturbances, so that NEP = NCE during equilibration. This is an interesting point that I have long wrestled with - in fact, how to equilibrate a model with disturbances because you can't just run the model for a set historical period with known disturbances, but have to run for hundreds of years to reach equilibration. The assumption itself that ecosystems can be in a state of equilibration is not true, but it is a necessary assumption to establish some baseline of initial conditions at some known starting year.

L275 Please explain why you expect that more mature forested in CONDENSED would have lower NEP. Also, please mention what period you are referring to. Finally, replace "mature forested" with "mature forests", if this is what you mean.

I have added two new references (Besnard et al., 2018; He et al., 2012), but the basic idea is that NEP reaches its peak in mid succession and eventually goes back toward 0 or slightly positive for mature trees. That is a good point about the time period, so I added the sentence "By the end of the century regrowing forests in the RESTART run will still be younger than those in CONDENSED run, and 85 years is not enough time to reach full equilibration in the model." (lines 358-360). Corrected typo.

L278 The CONDENSED run has been re-equilibrated to the environmental conditions of 20152045. Is that not the main reason why vegetation carbon is 16% larger compared to the RESTART simulation? Also, I don't recommend using the term bias here if you are not comparing against observation-based reference data (here and elsewhere).

Addressed this problem with the rerun of the CONDENSED experiment. Changed to "The larger values ..." (line 362).

L280 Explain exactly what you mean by "fixes most of the problem".

Changed to "lowers the vegetation carbon so that it is close to that of using the full cohorts" (lines 365-366).

L364 A difference of 1.0% or 1.8% does not seem very large and may not even be statistically significant.

Good point. I looked at the last 30 years, and did a t-test at the 0.05 confidence level – turns out the CONDENSED rerun is not different but the TEMRESTART is, so changed sentence to: "The soil moisture of the CONDENSED run over the last 30 years is not

**statistically different from RESTART, while the TEMRESTART is 1.8% higher during that time period (Fig. 12a)" (line 458-461).**

L391 I believe it is Chapin "III" et al. Also, this statement seems a little vague to me. Please explain what you mean.

**Fixed, and added "such that more mature trees have higher amounts of vegetation carbon" (lines 489-490).**

L411 Please mention what simulation you are referring to.

**Added, "from the HISTORICAL run" (line 510).**

L433 One reason why the modelled NEE IAV is smaller than observed may be related to the fact that your model does not represent mortality. That may be worth mentioning here as well.

I was actually trying to explain that the model does get within the correct range of measured IAV, and instead trying to point out that the entire time series shifts. Of course, if the values are still within the IAV, there are not significant changes. But I thought it still interesting to point out that the IAV did not change, only the means. If you feel this sentence should be removed, I can still do that.

L481 I assume that "models of the future" refer to models that that project future changes in vegetation dynamics? Please rephrase.

**Changed to "models simulating the future" (lines 612-613).**

L519 To decide whether a simulation is more realistic, you would need to evaluate your model output against some kind of observation-based reference data.

**Changed to "achieve a result more consistent with a detailed representation of land-use cohorts. (lines 650-651)"**

**Tables**

Table 1: Add information on how CO2, climate, and LULCC are treated in each experiment.

**Done**

**Figures**

Figure01 The time axis covers the period 1750-2014. Why does the caption say that the curves corresponds to the HISTORICAL and RESTART run, if the RESTART run starts in 2015?

**Sorry, my mistake - it's just the HISTORICAL run.**

S1b Please make larger to enhance readability.

**Done**

**Comment on bg-2022-208**

Anonymous Referee #2

- Referee comment on "Effect of land use legacy on the future carbon sink for the conterminous U.S." by Benjamin Seth Felzer, Biogeosciences Discuss., https://doi.org/10.5194/bg-2022-208-RC2, 2022
- This work examined how the initial state of the land surface affects simulated carbon cycles under an expected climate change. This is a kind of sensitivity analysis of a model because no systematic evaluation was conducted using observation-based data. Still, this work clarifies the importance of the initial state on simulations of transient changes in carbon pools and flux. I evaluate this model can be published in the BG if the author appropriately addresses the following issues.

**Major issues**

(1)This work evaluated results from only one model, "TEM-HYDRO2". Hence results are very model-specific. The author needs to discuss, at least, to what extent the finding in this work can be generalized.

This is an interesting point. After reviewing the full suite of TRENDY models, which serves as a nice intercomparison of leading edge DGVMs, I add that none of them actually include forest demography, though they do include changes in fractional land cover based on the LUC dataset, as well as conversion and product fluxes. I mention that Shiavlokova et al. (2009) does include forest demography using a tiling approach. In response to this comment, I have added a new experiment, HISTCOND, in which I condense all the cohorts in a transient historical run so that it is somewhat analogous to what many of the other models are doing. This (and another new run referenced above as HISTCONST) are now in Figure 2, the Methods, Results, and Discussion. The key paragraph addressing this point in the Discussion is (lines 549-562):

Most other terrestrial ecosystem models do not include the effect of forest demography. The Dynamic Global Vegetation Models (DGVM) included in Trends in Net Land-Atmosphere Exchange (TRENDY-v2) (Li et al., 2017) mostly include annual changes in PFTs to represent LULCC. They include the conversion and product fluxes resulting from these changes, and often include the effects of mortality and regrowth within existing grids, but do not incorporate the effects of forest regrowth due to LULCC. Two of the models (VISIT and JSBACH) (Kato et al., 2013; Reick et al., 2013) include elaborate

methods of applying the LULCC transition matrices to ensure the correct redistribution of PFTs and correct carbon fluxes. Shevliokova et al. (2009) does use a tiling approach to consider forest stand age and reduce the large number of cohorts used here. The HISTCOND run was designed specifically to explore the effects of forest demography by trying to emulate the effect of just redistributing annual land-use fractions. As seen in the results, it does substantially overestimate the carbon stocks and underestimate the NEP compared to the run that includes the full effects of forest demography.

(2)Before applying the future climatic conditions, the CONDENSED run was first equilibrated by repeatedly inputting projected climate during 2016-2045 (Line 210). Why were future climatic conditions employed here? To let the model equilibrate at the year 2015 (as is indicated in table 1), climatic data during the last few decades before 2015 would be simply employed here.

**Agreed – I reran the CONDENSE run to equilibrate from 1986-2014. See comments responding to the other reviewer, who had the same comment (lines 258-259).**

(3)The manuscript lacks descriptions of the nature of the LULC data and how the model implemented it.

I too would have preferred a more thorough explanation of our cohort approach, which I originally had in the manuscript, but was told I had to remove it because it was self-plagiarizing. So, instead I referred to a paper with the detailed description (see below, as in the manuscript). However, following comment from the editor, I did add extra material to describe the cohort approach. The first paragraph of the methods references the LUH2 dataset.

A cohort approach is developed to convert a dataset of land use transitions (Hurtt et al. (2011; 2020) to annual cohorts of land use and land cover change (Hayes et al., 2011; Lu et al., 2015), whose purpose is to retain the soil characteristics of the cohort from which disturbance occurred and maintain appropriate growth and stand age of newly developed cohorts (Fig. S2a). This approach involves first using the LUH2 dataset to establish the fractional land cover type at the starting year of 1750. The primary and secondary vegetation are replaced with their potential vegetation values (as described in (Raich et al., 1991)), while other managed lands include croplands, pasturelands, and urban, with the multiple types of crops and pastures combined into single values for each, respectively. Disturbances (including timber harvest) involve the creation of new cohorts, with the corresponding area adjusted from the original cohort. Therefore, soil nutrients and forest stand age are tracked separately for each disturbance. The output are then area-weighted for each of the cohorts. Since this approach tracks each cohort separately, it is possible to end up with thousands of cohorts for a single grid cell by 2014. A complete description of this approach can be found in Felzer and Jiang (2018). (lines 131-140)

Minor issues

(4)Please unify model names. The present manuscript utilizes both "TEM-HYDRO2" and "TEM-Hydro" for the same model.

**Ok, I removed the acronym TEM-HYDRO2.**

(5)Words on some of the figures are too small. Please magnify them.

I have redone all the figures with Time New Roman 12 (and 14 for the titles). This was particularly a problem for the legends, which were in 10-point font. When I copy and paste them into the document, and then resize them, the text may have gotten smaller. For the final version, I will be providing high quality files that should not have this problem.

(6)Line 80

What is the "FIA" stands for?

Added "Forest Inventory Analysis" (lines 81-82).

(7) Line 201 "Future CO2 data are taken from Meinshausen et al.(2020)." Is this data correspond to the RCP8.5? The author needs to inform about it.(8)

Added "RCP8.5" but also fixed the reference to Meinshausen et al. (2011), so now reads: . Future RCP8.5 CO2 data are taken from CMIP5 recommendations (Meinshausen et al., 2011) (lines 234-235).

(9)Line 212-214

Please clarify the difference between NCE and NEE in this work.

Both reviewers had same comment – added sentence: "NCE is the NEP plus carbon lost through land-use conversion or by decomposition of agricultural or timber harvest products" (lines 261-263).

(9) Line 342 "the amount available"

Does it mean "the amount of available inorganic nitrogen"?

Yes, added "inorganic" (line 445).

Review

(10) Line 385 "NE U.S." What it stands for?

(11) Lines 423-424

Values here correspond to NEP or NCE?

For NEP, so I clarified that, though the standard deviations are very close for NCE as well (line 523).